# ON ERROR PROPAGATION OF DIFFUSION MODELS

**Yangming Li, Mihaela van der Schaar**
Department of Applied Mathematics and Theoretical Physics
University of Cambridge
`yl874@cam.ac.uk`

## ABSTRACT

Although diffusion models (DMs) have shown promising performances in a number of tasks (e.g., speech synthesis and image generation), they might suffer from *error propagation* because of their sequential structure. However, this is not certain because some sequential models, such as Conditional Random Field (CRF), are free from this problem. To address this issue, we develop a theoretical framework to mathematically formulate *error propagation* in the architecture of DMs, The framework contains three elements, including *modular error*, *cumulative error*, and *propagation equation*. The modular and cumulative errors are related by the equation, which interprets that DMs are indeed affected by *error propagation*. Our theoretical study also suggests that the *cumulative error* is closely related to the generation quality of DMs. Based on this finding, we apply the *cumulative error* as a regularization term to reduce *error propagation*. Because the term is computationally intractable, we derive its upper bound and design a bootstrap algorithm to efficiently estimate the bound for optimization. We have conducted extensive experiments on multiple image datasets, showing that our proposed regularization reduces *error propagation*, significantly improves vanilla DMs, and outperforms previous baselines.

## 1 INTRODUCTION

**DMs potentially suffer from error propagation.** While diffusion models (DMs) have shown impressive performances in a number of tasks (e.g., image synthesis (Rombach et al., 2022), speech processing (Kong et al., 2021) and natural language generation (Li et al., 2022)), they might still suffer from *error propagation* (Motter & Lai, 2002), a classical problem in engineering practices (e.g., communication networks) (Fu et al., 2020; Motter & Lai, 2002; Crucitti et al., 2004). The problem mainly affects some chain models that consist of many end-to-end connected modules. For those models, the output error from one module will spread to subsequent modules such that errors accumulate along the chain. Since DMs are exactly of a chain structure and have a large number of modules (e.g., 1000 for DDPM (Ho et al., 2020)), the effect of *error propagation* on diffusion models is potentially notable and worth a careful study.

**Current works lack reliable explanations.** Some recent works (Ning et al., 2023; Li et al., 2023; Daras et al., 2023) have noticed this problem and named it as *exposure bias* or *sampling drift*. However, those works claim that *error propagation* happens to DMs simply because the models are of a cascade structure. In fact, many sequential models (e.g., CRF (Lafferty et al., 2001) and CTC (Graves et al., 2006)) are free from the problem. Therefore, a solid explanation is expected to answer whether(or even why) *error propagation* impacts on DMs.

**Our theory for the error propagation of DMs.** One main focus of this work is to develop a theoretical framework for analyzing the *error propagation* of DMs. With this framework, we can clearly understand how this problem is mathematically formulated in the architecture of DMs and easily see whether it has a significant impact on the models.

Our framework contains three elements: *modular error*, *cumulative error*, and *propagation equation*. The first two elements respectively measure the prediction error of one single module and the accumulated error of multiple modules, while the last one tells how these errors are related. We

first properly define the errors and then derive the equation, which will be applied to explain why *error propagation* happens to DMs. We will see that *it is a term called amplification factor in the propagation equation that determines whether error accumulation happens*. Besides the theoretical study, we also perform empirical experiments to verify whether our theory applies to reality.

**Why and how to reduce error propagation.** In our theoretical study, we will see that the *cumulative error* actually reflects the generation quality of DMs. Hence, if *error propagation* is very serious in DMs, then we can improve the generation performances by reducing it.

A simple way to reducing error propagation is to treat the *cumulative error* as a regularization term and directly minimize it at training time. However, we will see that this error is computationally infeasible in terms of both density estimation and inefficient backward process. Therefore, we first introduce an upper bound of the error, which avoids estimating densities. Then, we design a bootstrap algorithm (which is inspired by TD learning (Tesauro et al., 1995; Osband et al., 2016)) to efficiently estimate the bound though with a certain bias.

**Contributions.** The contributions of our paper are as follows:

- We develop a theoretical framework for analyzing the *error propagation* of DMs, which explains whether the problem affects DMs and why it should be solved. Compared with our framework, previous explanations are just based on an unreliable intuition;

- In light of our theoretical study, we introduce a regularization term to reduce *error propagation*. Since the term is computationally infeasible, we derive its upper bound and design a bootstrap algorithm to efficiently estimate the bound for optimization;

- We have conducted extensive experiments on multiple image datasets, demonstrating that our proposed regularization reduces *error propagation*, significantly improves the performances of vanilla DMs, and outperforms previous baselines.

## 2 BACKGROUND: DISCRETE-TIME DMS

In this section, we briefly review the mainstream architecture of DMs (i.e., DDPM). The notations and terminologies introduced below will be used in our subsequent sections.

A DM consists of two Markov chains of $T$ steps. One is the forward process, which incrementally adds Gaussian noises into real sample $\mathbf{x}_0 \sim q(\mathbf{x}_0)$, a $K$-dimensional vector. In this process, a sequence of latent variables $\mathbf{x}_{1:T} = [\mathbf{x}_1, \mathbf{x}_2, \cdots, \mathbf{x}_T]$ are generated in order and the last one $\mathbf{x}_T$ will approximately follow a standard Gaussian:

$$q(\mathbf{x}_{1:T} \mid \mathbf{x}_0) = \prod_{t=1}^{T} q(\mathbf{x}_t \mid \mathbf{x}_{t-1}), \quad q(\mathbf{x}_t \mid \mathbf{x}_{t-1}) = \mathcal{N}(\mathbf{x}_t; \sqrt{1-\beta_t}\mathbf{x}_{t-1}, \beta_t \mathbf{I}), \quad (1)$$

where $\mathcal{N}$ denotes a Gaussian distribution, $\mathbf{I}$ is an identity matrix, and $\beta_t, 1 \leqslant t \leqslant T$ represents a predefined variance schedule.

The other is the backward process, a reverse version of the forward process, which first draws an initial sample $\mathbf{x}_T$ from standard Gaussian $p(\mathbf{x}_T) = \mathcal{N}(\mathbf{0}, \mathbf{I})$ and then gradually denoises it into a sequence of latent variables $\mathbf{x}_{T-1:0} = [\mathbf{x}_{T-1}, \mathbf{x}_{T-2}, \cdots, \mathbf{x}_0]$ in reverse order:

$$p_\theta(\mathbf{x}_{T:0}) = p(\mathbf{x}_T) \prod_{t=T}^{1} p_\theta(\mathbf{x}_{t-1} \mid \mathbf{x}_t), \quad p_\theta(\mathbf{x}_{t-1} \mid \mathbf{x}_t) = \mathcal{N}(\mathbf{x}_{t-1}; \boldsymbol{\mu}_\theta(\mathbf{x}_t, t), \sigma_t \mathbf{I}), \quad (2)$$

where $p_\theta(\mathbf{x}_{t-1} \mid \mathbf{x}_t)$ is a denoising module with the parameter $\theta$ shared across different iterations, and $\sigma_t \mathbf{I}$ is a predefined covariance matrix.

Since the negative log-likelihood $\mathbb{E}[-\log p_\theta(\mathbf{x}_0)]$ is computationally intractable, common practices apply Jensen's inequality to derive its upper bound for optimization:

$$\mathbb{E}_{\mathbf{x}_0 \sim q(\mathbf{x}_0)}[-\log p_\theta(\mathbf{x}_0)] \leqslant \mathbb{E}_q[D_{\mathrm{KL}}(q(\mathbf{x}_T \mid \mathbf{x}_0) \,\|\, p(\mathbf{x}_T))] + \mathbb{E}_q[-\log p_\theta(\mathbf{x}_0 \mid \mathbf{x}_1)]$$
$$+ \sum_{1 \leqslant t < T} \mathbb{E}_q[D_{\mathrm{KL}}(q(\mathbf{x}_{t-1} \mid \mathbf{x}_t, \mathbf{x}_0) \,\|\, p_\theta(\mathbf{x}_{t-1} \mid \mathbf{x}_t))] = \mathcal{L}^{\mathrm{nll}} \qquad , \quad (3)$$

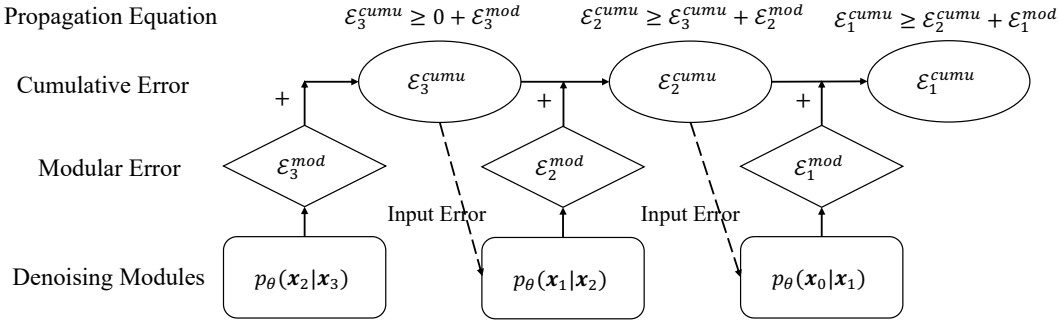

Figure 1: A toy example (with $T = 3$) to show our theoretical framework for the *error propagation* of diffusion models. We use dash lines to indicate that the impact of *cumulative error* $\mathcal{E}_{t+1}^{\mathrm{cumu}}$ on denoising module $p_\theta(\mathbf{x}_{t-1} \mid \mathbf{x}_t)$ is defined at the distributional (not sample) level.

where $D_{\mathrm{KL}}$ denotes the KL divergence and each term has an analytic form for feasible computation. Ho et al. (2020) further applied some reparameterization tricks to the loss $\mathcal{L}^{\mathrm{nll}}$ to reduce its estimation variance. For example, the mean $\boldsymbol{\mu}_\theta$ is formulated by

$$\boldsymbol{\mu}_\theta(\mathbf{x}_t, t) = \frac{1}{\sqrt{\alpha_t}}\left(\mathbf{x}_t - \frac{\beta_t}{\sqrt{1 - \bar{\alpha}_t}}\boldsymbol{\epsilon}_\theta(\mathbf{x}_t, t)\right), \tag{4}$$

in which $\alpha_t = 1 - \beta_t$, $\bar{\alpha}_t = \prod_{t'=1}^{t}\alpha_{t'}$, and $\boldsymbol{\epsilon}_\theta$ is parameterized by a neural network. Under this scheme, the loss $\mathcal{L}^{\mathrm{nll}}$ can be simplified as

$$\mathcal{L}^{\mathrm{nll}} = \sum_{t=1}^{T}\underbrace{\mathbb{E}_{\mathbf{x}_0 \sim q(\mathbf{x}_0), \boldsymbol{\epsilon} \sim \mathcal{N}(\mathbf{0}, \mathbf{I})}\left[\|\boldsymbol{\epsilon} - \boldsymbol{\epsilon}_\theta(\sqrt{\bar{\alpha}_t}\mathbf{x}_0 + \sqrt{1 - \bar{\alpha}_t}\boldsymbol{\epsilon}, t)\|^2\right]}_{\mathcal{L}_t^{\mathrm{nll}}}, \tag{5}$$

where the neural network $\boldsymbol{\epsilon}_\theta$ is tasked to fit Gaussian noise $\boldsymbol{\epsilon}$.

## 3 THEORETICAL STUDY

In this section, we first introduce our theoretical framework and discuss why current explanations (for why *error propagation* happens to DMs) are not reliable. Then, we seek to define (or derive) the elements of our framework and apply them to give a solid explanation.

### 3.1 ANALYSIS FRAMEWORK

**Elements of the framework.** Fig. 1 illustrates a preview of our framework. We can see from the preview that we have to find the following three elements from DMs:

- *Modular error* $\mathcal{E}_t^{\mathrm{mod}}$, which measures how accurately every module maps its input to the output. It only relies on the module at iteration $t$ and is independent of the others. Intuitively, the error is non-negative: $\mathcal{E}_t^{\mathrm{mod}} \geqslant 0, \forall t \in [1, T]$;

- *Cumulative error* $\mathcal{E}_t^{\mathrm{cumu}}$, which is also non-negative, measuring the amount of error that can be accumulated for sequentially running the first $T - t + 1$ denoising modules. Unlike the *modular error*, its value is influenced by more than one modules;

- *Propagation equation* $f(\mathcal{E}_1^{\mathrm{cumu}}, \mathcal{E}_2^{\mathrm{cumu}}, \cdots, \mathcal{E}_T^{\mathrm{cumu}}, \mathcal{E}_1^{\mathrm{mod}}, \mathcal{E}_2^{\mathrm{mod}}, \cdots, \mathcal{E}_T^{\mathrm{mod}}) = 0$, which tells how the modular and cumulative errors of different iterations are related. It is generally very hard to find such an equation because the model is non-linear.

**Interpretation of the propagation equation.** In a DM, the input error to every module $p_\theta(\mathbf{x}_{t-1} \mid \mathbf{x}_t), t \in [1, T]$ can be regarded as $\mathcal{E}_{t+1}^{\mathrm{cumu}}$. Since module $p_\theta(\mathbf{x}_{t-1} \mid \mathbf{x}_t)$ is parameterized by a non-linear neural networks, it is not certain whether its input error $\mathcal{E}_{t+1}^{\mathrm{cumu}}$ is magnified or reduced in the

error contained in its output: $\mathcal{E}_t^{\text{cumu}}$. Therefore, we expect the *propagation equation* to include or just take the form of the following equation:

$$\mathcal{E}_t^{\text{cumu}} - \mathcal{E}_t^{\text{mod}} = \mu_t \mathcal{E}_{t+1}^{\text{cumu}}, \tag{6}$$

which specifies how much input error $\mathcal{E}_{t+1}^{\text{cumu}}$ that the module $p_\theta(\mathbf{x}_{t-1} \mid \mathbf{x}_t)$ spreads to the subsequent modules. For example, $\mu_t < 1$ means the model can discount the input *cumulative error* $\mathcal{E}_{t+1}^{\text{cumu}}$, which contributes to avoid error accumulation. We also term $\mu_t$ as the *amplification factor*, which satisfies $\mu_t \geqslant 0$ because the *cumulative error* $\mathcal{E}_t^{\text{cumu}}$ at least contains the prediction error of the denoising module $p_\theta(\mathbf{x}_{t-1} \mid \mathbf{x}_t)$.

With this equation, *if we have $\mu_t \geqslant 1, \forall t \in [1, T]$, then we can assert that the DM is affected by error propagation* because in that case every module $p_\theta(\mathbf{x}_{t-1} \mid \mathbf{x}_t)$ either fully spreads or even magnifies its input error to the subsequent modules.

**Why current explanations are not solid.** Current works (Ning et al., 2023; Li et al., 2023; Daras et al., 2023) claim that DMs are affected by *error propagation* just because they are of a chain structure. Based on our theoretical framework, we can see that this intuition is not reliable. For example, if $\mu_t = 0, \forall t \in [1, T]$, every module $p_\theta(\mathbf{x}_{t-1} \mid \mathbf{x}_t)$ of the DM will fully reduce its input *cumulative error* $\mathcal{E}_{t+1}^{\text{cumu}}$, which avoids *error propagation*.

## 3.2 ERROR DEFINITIONS

In this part, we formally define the modular and cumulative errors, which respectively measure the prediction error of one module and the accumulated errors of multiple successive modules.

**Derivation of the modular error.** In common practices (Ho et al., 2020; Song et al., 2021b), every denoising module in a diffusion model $p_\theta(\mathbf{x}_{t-1} \mid \mathbf{x}_t), t \in [1, T]$ is optimized to exactly match the posterior probability $q(\mathbf{x}_{t-1} \mid \mathbf{x}_t)$. Considering this fact and the forms of loss terms in Eq. (3), we apply KL divergence to measure the discrepancy between those two conditional distributions: $D_{\text{KL}}(p_\theta(\cdot) \parallel q(\cdot))$, and treat it as the *modular error*. To further get ride of the dependence on variable $\mathbf{x}_t$, we apply an expectation operation $\mathbb{E}_{\mathbf{x}_t \sim p_\theta(\mathbf{x}_t)}$ to the error.

**Definition 3.1** (Modular Error). For the forward process as defined in Eq. (1) and the backward process as defined in Eq. (2), the *modular error* $\mathcal{E}_t^{\text{mod}}$ of every denoising module $p_\theta(\mathbf{x}_{t-1} \mid \mathbf{x}_t), t \in [1, T]$ in a diffusion model is measured as

$$\mathcal{E}_t^{\text{mod}} = \mathbb{E}_{\mathbf{x}_t \sim p_\theta(\mathbf{x}_t)}[D_{\text{KL}}(p_\theta(\mathbf{x}_{t-1} \mid \mathbf{x}_t) \parallel q(\mathbf{x}_{t-1} \mid \mathbf{x}_t))]. \tag{7}$$

Inequality $\mathcal{E}_t^{\text{mod}} \geqslant 0$ always holds since KL divergence is non-negative.

**Derivation of the cumulative error.** Considering the fact (Ho et al., 2020): $q(\mathbf{x}_t \mid \mathbf{x}_0) = \mathcal{N}(\mathbf{x}_t; \sqrt{\bar{\alpha}_t}\mathbf{x}_0, (1 - \bar{\alpha}_t)\mathbf{I})$, we can reparameterize the variable $\mathbf{x}_t$ as $\sqrt{\bar{\alpha}_t}\mathbf{x}_0 + \sqrt{1 - \bar{\alpha}_t}\boldsymbol{\epsilon}, \boldsymbol{\epsilon} \sim \mathcal{N}(\mathbf{0}, \mathbf{I})$. Therefore, the loss term $\mathcal{L}_t^{\text{nll}}$ in Eq. (5) can be reformulated as

$$\mathcal{L}_t^{\text{nll}} = \mathbb{E}_{\mathbf{x}_0 \sim q(\mathbf{x}_0), \mathbf{x}_t \sim q(\mathbf{x}_t \mid \mathbf{x}_0)}\left[\left\|\frac{\mathbf{x}_t - \sqrt{\bar{\alpha}_t}\mathbf{x}_0}{\sqrt{1 - \bar{\alpha}_t}} - \boldsymbol{\epsilon}_\theta(\mathbf{x}_t, t)\right\|^2\right],$$

From this equality, we can see that the input $\mathbf{x}_t$ to the module $p_\theta(\mathbf{x}_{t-1} \mid \mathbf{x}_t)$ at training time follows a fixed distribution that is specified by the forward process $q(\mathbf{x}_{T:0})$:

$$\int_{\mathbf{x}_0} q(\mathbf{x}_0)q(\mathbf{x}_t \mid \mathbf{x}_0)d\mathbf{x}_0 = \int_{\mathbf{x}_0} q(\mathbf{x}_t, \mathbf{x}_0) = q(\mathbf{x}_t), \tag{8}$$

which is inconsistent with the input distribution that the module $p_\theta(\mathbf{x}_{t-1} \mid \mathbf{x}_t)$ receives from its previous denoising module $\{p_\theta(\mathbf{x}_{t'-1} \mid \mathbf{x}_{t'}) \mid t' \in [t + 1, T]\}$ during evaluation:

$$\int_{\mathbf{x}_{T:t+1}} p(\mathbf{x}_T) \prod_{i=T}^{t+1} p_\theta(\mathbf{x}_{i-1} \mid \mathbf{x}_i)d\mathbf{x}_{T:t+1} = \int_{\mathbf{x}_{T:t+1}} p_\theta(\mathbf{x}_t, \mathbf{x}_{T:t+1})d\mathbf{x}_{T:t+1} = p_\theta(\mathbf{x}_t). \tag{9}$$

The discrepancy between these two distributions, $q(\mathbf{x}_t)$ and $p_\theta(\mathbf{x}_t)$, is essentially the input error to the denoising module $p_\theta(\mathbf{x}_{t-1} \mid \mathbf{x}_t)$. Like the *modular error*, we also apply KL divergence to measure this discrepancy as $D_{\text{KL}}(p_\theta(\mathbf{x}_t) \parallel q(\mathbf{x}_t))$. Because the input error of module $p_\theta(\mathbf{x}_{t-1} \mid \mathbf{x}_t)$ can be regarded as the accumulated output errors of its all previous modules, we resort to that input error to define the *cumulative error*.

**Definition 3.2** (Cumulative Error). Given the forward and backward processes that are respectively defined in Eq. (1) and Eq. (2), the *cumulative error* $\mathcal{E}_t^{\mathrm{cumu}}$ of the diffusion model at iteration $t \in [1, T]$ (caused by the first $T - t + 1$ modules) is measured as

$$\mathcal{E}_t^{\mathrm{cumu}} = D_{\mathrm{KL}}(p_\theta(\mathbf{x}_{t-1}) \,\|\, q(\mathbf{x}_{t-1})). \tag{10}$$

Inequality $\mathcal{E}_t^{\mathrm{cumu}} \geqslant 0$ always holds because KL divergence is non-negative.

*Remark* 3.1. *The error $\mathcal{E}_t^{\mathrm{cumu}}$ in fact reflects the performance of DMs in data generation.* For instance, when $t = 1$, the *cumulative error* $\mathcal{E}_1^{\mathrm{cumu}} = D_{\mathrm{KL}}(p_\theta(\mathbf{x}_0) \,\|\, q(\mathbf{x}_0))$ indicates whether the generated samples are distributionally consistent with real data. Therefore, *a method to reducing error propagation is expected to improve the performance of DMs.*

*Remark* 3.2. In Appendix A, we prove that $\mathcal{E}_t^{\mathrm{cumu}} = 0$ is achievable in an ideal case: the diffusion model is *perfectly optimized* (i.e., $p_\theta(\mathbf{x}_{t'-1} \mid \mathbf{x}_{t'}) = q(\mathbf{x}_{t'-1} \mid \mathbf{x}_{t'}), \forall t' \in [1, T]$).

### 3.3 PROPAGATION EQUATION

The propagation equation describes how the modular and cumulative errors of different iterations are related. We derive that equation for the diffusion models as below.

**Theorem 3.1** (Propagation Equation). *For the forward and backward processes respectively defined in Eq. (1) and Eq. (2), suppose that the output of neural network $\boldsymbol{\epsilon}_\theta(\cdot)$ (as defined in Eq. (4)) follows a standard Gaussian regardless of input distributions and the entropy of distribution $p_\theta(\mathbf{x}_t)$ is non-increasing with decreasing iteration $t$, then the following inequality holds:*

$$\mathcal{E}_t^{\mathrm{cumu}} \geqslant \mathcal{E}_{t+1}^{\mathrm{cumu}} + \mathcal{E}_t^{\mathrm{mod}}, \tag{11}$$

*where $t \in [1, T]$ and we specially set $\mathcal{E}_{T+1} = 0$.*

*Remark* 3.3. The theorem is based on two main intuitive assumptions: 1) From Eq. (5), we can see that neural network $\boldsymbol{\epsilon}_\theta$ aims to fit Gaussian noise $\boldsymbol{\epsilon}$ and is shared by all denoising iterations, which means it takes the input from various distributions $[q(\mathbf{x}_T), q(\mathbf{x}_{T-1}), \cdots, q(\mathbf{x}_1)]$. Therefore, it's reasonable to assume that the output of neural network $\boldsymbol{\epsilon}_\theta$ follows a standard Gaussian distribution, independent of the input distribution; 2) The backward process is designed to incrementally denoise Gaussian noise $\mathbf{x}_T \sim \mathcal{N}(\mathbf{0}, \mathbf{I})$ into real sample $\mathbf{x}_0$. Ideally, the uncertainty (i.e., noise level) of distribution $p_\theta(\mathbf{x}_t)$ gradually decreases in that denoising process. From this view, it makes sense to assume that differential entropy $H_{p_\theta}(\mathbf{x}_t)$ reduces with decreasing iteration $t$.

*Proof.* We provide a complete proof to this theorem in Appendix C. □

**Are DMs affected by error propagation?** By comparing Eq. (11) and Eq. (6), we can see that $\mu_t \geqslant 1, \forall t \in [1, T]$. Based on our discussion in Sec. 3.1, we can assert that *error propagation* happens to standard DMs. The basic idea: $\mu_t \geqslant 1$ means that the module $p_\theta(\mathbf{x}_{t-1} \mid \mathbf{x}_t)$ at least fully spreads the input *cumulative error* $\mathcal{E}_{t+1}^{\mathrm{cumu}}$ to its subsequent modules.

## 4 EMPIRICAL STUDY: CUMULATIVE ERROR ESTIMATION

Besides the theoretical study, it is also important to empirically verify whether diffusion models are affected by error propagation. Hence, we aim to numerically compute the cumulative error $\mathcal{E}_t^{\mathrm{cumu}}$ and show how it changes with decreasing iteration $t \in [1, T]$. By doing so, we can also verify whether our theory (e.g., propgation equation: Eq. (11)) applies to reality.

Because the marginal distributions $p_\theta(\mathbf{x}_{t-1}), q(\mathbf{x}_{t-1})$ have no closed-form solutions, the KL divergence between them (i.e., the cumulative error $\mathcal{E}_t^{\mathrm{cumu}}$) is computationally infeasible. To address this problem, we adopt the maximum mean discrepancy (MMD) (Gretton et al., 2012) to estimate the error $\mathcal{E}_t^{\mathrm{cumu}}$ and prove that it is in fact tightly bounded by MMD.

With MMD, we can define another type of cumulative error $\mathcal{D}_t^{\mathrm{cumu}}, t \in [1, T]$:

$$\mathcal{D}_t^{\mathrm{cumu}} = \left\| \mathbb{E}_{\mathbf{x}_{t-1} \sim p_\theta(\mathbf{x}_{t-1})}[\phi(\mathbf{x}_{t-1})] - \mathbb{E}_{\mathbf{x}_{t-1} \sim q(\mathbf{x}_{t-1})}[\phi(\mathbf{x}_{t-1})] \right\|_{\mathcal{H}}^2, \tag{12}$$

which also measures the gap between $p_\theta(\mathbf{x}_{t-1})$ and $q(\mathbf{x}_{t-1})$. Here $\mathcal{H}$ denotes a reproducing kernel Hilbert space and $\phi : \mathbb{R}^K \to \mathcal{H}$ is a feature map. Following common practices (Dziugaite et al.,

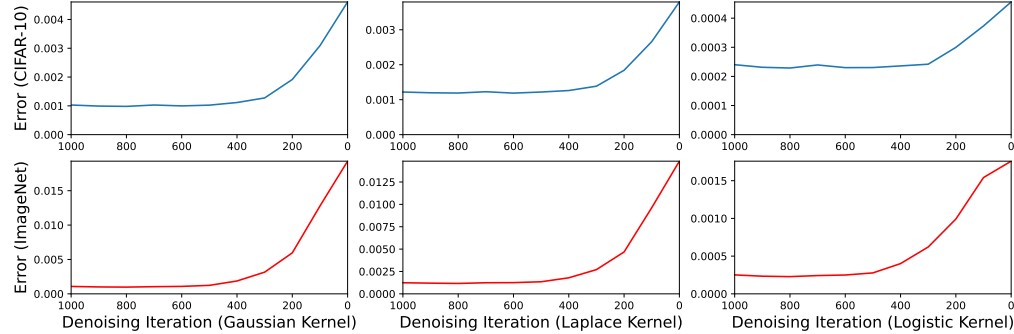

Figure 2: Uptrend dynamics of the MMD error $\mathcal{D}_t^{\text{cumu}}$ w.r.t. decreasing iteration $t$. The cumulative error $\mathcal{E}_t^{\text{cumu}}$ might show similar behaviors since it is tightly bounded by the MMD error.

2015), we adopt an unbiased estimate of the error $\mathcal{D}_t^{\text{cumu}}$:

$$
\begin{aligned}
\mathcal{D}_t^{\text{cumu}} \approx \frac{1}{N^2} \sum_{1 \leqslant i,j \leqslant N} \mathcal{K}(\mathbf{x}_{t-1}^{\text{back},i}, \mathbf{x}_{t-1}^{\text{back},j}) + \frac{1}{M^2} \sum_{1 \leqslant i,j \leqslant M} \mathcal{K}(\mathbf{x}_{t-1}^{\text{forw},i}, \mathbf{x}_{t-1}^{\text{forw},j}) \\
- \frac{2}{NM} \sum_{1 \leqslant i \leqslant N, 1 \leqslant j \leqslant M} \mathcal{K}(\mathbf{x}_{t-1}^{\text{back},i}, \mathbf{x}_{t-1}^{\text{forw},j})
\end{aligned}
\tag{13}
$$

where $\mathbf{x}_{t-1}^{\text{back},i} \sim p_\theta(\mathbf{x}_{t-1}), 1 \leqslant i \leqslant N$ is performed with Eq. (2), $\mathbf{x}_{t-1}^{\text{forw},j} \sim q(\mathbf{x}_{t-1}), 1 \leqslant j \leqslant M$ is done by Eq. (1), and $\mathcal{K}(\cdot)$ is the kernel function to simplify the inner products among $\{\phi(\mathbf{x}_{t-1}^{\text{back},1}), \phi(\mathbf{x}_{t-1}^{\text{back},2}), \cdots, \phi(\mathbf{x}_{t-1}^{\text{back},N}), \phi(\mathbf{x}_{t-1}^{\text{forw},1}), \phi(\mathbf{x}_{t-1}^{\text{forw},2}), \cdots, \phi(\mathbf{x}_{t-1}^{\text{forw},M})\}$.

Importantly, we show in the following that the cumulative error $\mathcal{E}_t^{\text{cumu}}$ is tightly bounded by the MMD error $\mathcal{D}_t^{\text{cumu}}$ from below and above.

**Proposition 4.1** (Bounds of the Cumulative Error). *Suppose that $\mathcal{H} \subseteq \mathcal{C}_\infty(\mathbb{R}^K)$ and the two continuous marginal distributions $p_\theta(\mathbf{x}_{t-1}), q(\mathbf{x}_{t-1})$ are non-zero everywhere, then the cumulative error $\mathcal{E}_t^{\text{cumu}}$ is linearly bounded by the MMD error $\mathcal{D}_t^{\text{cumu}}$ from below and above:*

$$
\frac{1}{4}\mathcal{D}_t^{\text{cumu}} \leqslant \mathcal{E}_t^{\text{cumu}} \leqslant \mathcal{D}_t^{\text{cumu}}.
\tag{14}
$$

*Here $\mathcal{C}_\infty(\mathbb{R}^K)$ is the set of all continuous functions (with finite uniform norms) over $\mathbb{R}^K$.*

*Proof.* A complete proof to this proposition is provided in Appendix E. $\square$

**Experiment setup and results.** We train standard diffusion models (Ho et al., 2020) on two datasets: CIFAR-10 ($32 \times 32$) (Krizhevsky et al., 2009) and ImageNet ($32 \times 32$) (Deng et al., 2009). Three different kernel functions (e.g., Laplace kernel) are adopted to compute the error $\mathcal{D}_t^{\text{cumu}}$ in terms of Eq. (13). Both $N$ and $M$ are set as 1000.

As shown in Fig. 2, the MMD error $\mathcal{D}_t^{\text{cumu}}$ rapidly grows with respect to decreasing iteration $t$ in all settings, implying that the *cumulative error* $\mathcal{E}_t^{\text{cumu}}$ also has similar trends. The results not only empirically verify that diffusion models are affected by error propagation, but also indicate that our theory (though built under some assumptions) applies to reality.

## 5  METHOD

According to Remark 3.1, the *cumulative error* in fact reflects the generation quality of DMs. Therefore, we might improve the performances of DMs through reducing *error propagation*. In light of this finding, we first introduce a basic version of our approach to reduce error propagation, which covers our core idea but is not practical in terms of running time. Then, we extend this approach to practical use through bootstrapping.

## 5.1 REGULARIZATION WITH CUMULATIVE ERRORS

An obvious way for reducing *error propagation* is to treat the *cumulative error* $\mathcal{E}_t^{\mathrm{cumu}}$ as a regularization term to optimize DMs. Since this term is computationally intractable as mentioned in Sec. 4, we adopt its upper bound $\mathcal{D}_t^{\mathrm{cumu}}$ based on Proposition 4.1):

$$\mathcal{L}_t^{\mathrm{reg}} = \mathcal{D}_t^{\mathrm{cumu}}, \quad \mathcal{L}^{\mathrm{reg}} = \sum_{t=0}^{T-1} w_t \mathcal{L}_t^{\mathrm{reg}}, \tag{15}$$

where $w_t \propto \exp(\rho * (T - t)), \rho \in \mathbb{R}^+$ and $\sum_{t=0}^{T-1} w_t = 1$. We exponentially schedule the weight $w_t$ because Fig. 2 suggests that error propagation is more severe as $t$ gets closer to 0. Term $\mathcal{L}_t^{\mathrm{reg}}$ is estimated by Eq. (13) in practice, with backward variable $\mathbf{x}_t^{\mathrm{back},i}$ denoised from a Gaussian noise via Eq. (2) and forward variable $\mathbf{x}_t^{\mathrm{forw},j}$ converted from a real sample with Eq. (1).

The new objective $\mathcal{L}^{\mathrm{reg}}$ is in addition to the main one $\mathcal{L}^{\mathrm{nll}}$ (previously defined in Eq. (5)) for regularizing the optimization of diffusion models. We linearly combine them as $\mathcal{L} = \lambda^{\mathrm{nll}} \mathcal{L}^{\mathrm{nll}} + \lambda^{\mathrm{reg}} \mathcal{L}^{\mathrm{reg}}$, where $\lambda^{\mathrm{nll}}, \lambda^{\mathrm{reg}} \in (0, 1)$ and $\lambda^{\mathrm{nll}} + \lambda^{\mathrm{reg}} = 1$, for joint optimization.

## 5.2 BOOTSTRAPPING FOR EFFICIENCY

A challenge of applying our approach in practice is the inefficient backward process. Specifically, to sample backward variable $\mathbf{x}_t^{\mathrm{back},i}$ for estimating loss $\mathcal{L}_t^{\mathrm{reg}}$, we have to iteratively apply a sequence of $T - t$ modules to cast a Gaussian noise into that variable.

Inspired by temporal difference (TD) learning (Tesauro et al., 1995), we bootstrap the computation of variable $\mathbf{x}_t^{\mathrm{back},i}$ for run-time efficiency. To this end, we first sample a time point $s > t$ from a uniform distribution $\mathcal{U}\{\min(t + L, T), t + 1\}$ and apply the forward process to estimate variable $\mathbf{x}_s^{\mathrm{back},i}$ in a possibly biased manner:

$$\widetilde{\mathbf{x}}_s^{\mathrm{back},i} = \sqrt{\bar{\alpha}_s}\mathbf{x}_0 + \sqrt{1 - \bar{\alpha}_s}\boldsymbol{\epsilon}, \mathbf{x}_0 \sim q(\mathbf{x}_0), \boldsymbol{\epsilon} \sim \mathcal{N}(\mathbf{0}, \mathbf{I}), \tag{16}$$

where operation $\sim q(\mathbf{x}_0)$ means to sample from training data and $L \ll T$ is a predefined sampling length. The above equation is based on a fact (Ho et al., 2020):

$$q(\mathbf{x}_t \mid \mathbf{x}_0) = \mathcal{N}(\mathbf{x}_t; \sqrt{\bar{\alpha}_t}\mathbf{x}_0, (1 - \bar{\alpha}_t)\mathbf{I}).$$

Then, we apply a chain of denoising modules $[p_\theta(\mathbf{x}_{s-1} \mid \mathbf{x}_s), p_\theta(\mathbf{x}_{s-2} \mid \mathbf{x}_{s-1}), \cdots, p_\theta(\mathbf{x}_t \mid \mathbf{x}_{t+1})]$ to iteratively update $\widetilde{\mathbf{x}}_s^{\mathrm{back},i}$ into $\widetilde{\mathbf{x}}_t^{\mathrm{back},i}$, which can be treated as an alternative to variable $\mathbf{x}_t^{\mathrm{back},i}$. Each update can be formulated as

$$\widetilde{\mathbf{x}}_{k-1}^{\mathrm{back},i} = \frac{1}{\sqrt{\alpha_k}}\left(\widetilde{\mathbf{x}}_k^{\mathrm{back},i} - \frac{\beta_k}{\sqrt{1 - \bar{\alpha}_k}}\boldsymbol{\epsilon}_\theta(\widetilde{\mathbf{x}}_k^{\mathrm{back},i}, k)\right) + \sigma_k\boldsymbol{\epsilon}, \boldsymbol{\epsilon} \sim \mathcal{N}(\mathbf{0}, \mathbf{I}), \tag{17}$$

where $t + 1 \leqslant k \leqslant s$. Finally, we apply the alternative $\widetilde{\mathbf{x}}_t^{\mathrm{back},i}$ for computing loss $\mathcal{L}_t^{\mathrm{reg}}$ in terms of Eq. (13), with at most $L \ll T$ runs of neural network $\boldsymbol{\epsilon}_\theta$.

For specifying the sampling length $L$, there is actually a trade-off between estimation accuracy and time efficiency For example, large $L$ reduces the negative impact from biased initialization $\widetilde{\mathbf{x}}_s^{\mathrm{back},i}$ but incurs a high time cost. We will study this problem in the experiment and also put the details of our training procedure in Appendix F. The source code of this work is publicly available at a personal repository: https://github.com/louisli321/epdm, and our lab repository: https://github.com/vanderschaarlab/epdm.

## 6 EXPERIMENTS

We have performed extensive experiments to verify the effectiveness of our proposed regularization. Besides special studies (e.g., effect of some hyper-parameter), our main experiments include: 1) By comparing with Fig. 2, we show that the *error propagation* is reduced after applying our regularization; 2) On three image datasets (CIFAR-10, CelebA, and ImageNet), our approach notably improves diffusion models and outperforms the baselines.

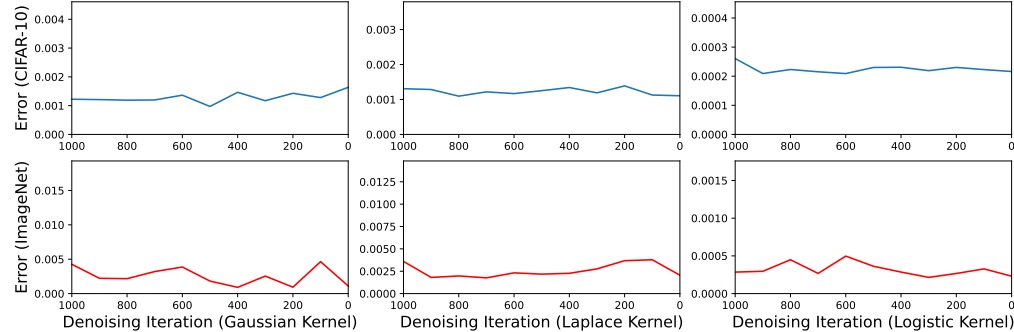

Figure 3: Re-estimated dynamics of the MMD error $\mathcal{D}_t^{\text{cumu}}$ with respect to decreasing iteration $t$ after applying our proposed regularization. These dynamics should be compared with those in Fig. 2, showing that we have well handled error propagation.

| Approach | CIFAR-10 | ImageNet | CelebA |
|---|---|---|---|
| ADM-IP Ning et al. (2023) | 3.25 | 2.72 | 1.31 |
| DDPM Ho et al. (2020) | 3.61 | 3.62 | 1.73 |
| DDPM w/ Consistent DM (Daras et al., 2023) | 3.31 | 3.16 | 1.38 |
| DDPM w/ FP-Diffusion (Lai et al., 2022) | 3.47 | 3.28 | 1.56 |
| DDPM w/ Our Proposed Regularization | **2.93** | **2.55** | **1.22** |

Table 1: FID scores of our model and baselines on different image datasets. The improvements of our approach over baselines are statistically significant with $p < 0.01$ under t-test.

## 6.1 SETTINGS

We conduct experiments on three image datasets: CIFAR-10 (Krizhevsky et al., 2009), ImageNet (Deng et al., 2009), and CelebA (Liu et al., 2015), with image shapes respectively as $32 \times 32$, $32 \times 32$, and $64 \times 64$. Following previous works (Ho et al., 2020; Dhariwal & Nichol, 2021), we adopt Frechet Inception Distance (FID) (Heusel et al., 2017) as the evaluation metric. The configuration of our model follows common practices, we adopt U-Net (Ronneberger et al., 2015) as the backbone and respectively set hyper-parameters $T, \sigma_t, L, \lambda^{\text{reg}}, \lambda^{\text{nll}}, \rho$ as $1000, \beta_t, 5, 0.2, 0.8, 0.003$. All our model run on $2 \sim 4$ Tesla V100 GPUs and are trained within two weeks.

**Baselines.** We compare our method with three baselines: ADM-IP Ning et al. (2023), Consistent DM (Daras et al., 2023), and FP-Diffusion (Lai et al., 2022). The first one is under the topic of exposure bias (which is related to our method) and the other two aim to regularize the DMs in terms of their inherent properties (e.g., Fokker-Planck Equation), which are less related to our method. The main idea of ADM-IP is to adds an extra Gaussian noise into the model input, which is not an ideal solution because this simple perturbation certainly can not simulate the complex errors at test time. Table 1 shows that our method significantly outperforms ADM-IP though ADM (Dhariwal & Nichol, 2021) is a stronger backbone than DDPM.

## 6.2 REDUCED ERROR PROPAGATION

To show the effect of our regularization $\mathcal{L}^{\text{reg}}$ to diffusion models, we re-estimate the dynamics of MMD error $\mathcal{D}_t^{\text{cumu}}$, with the same experiment setup of datasets and kernel functions as our empirical study in Sec. 4. The only difference is that the input distribution to module $p_\theta(\mathbf{x}_{t-1} \mid \mathbf{x}_t)$ at training time is no longer $q(\mathbf{x}_t)$, but we can still correctly estimate error $\mathcal{D}_t^{\text{cumu}}$ with Eq. (13) by re-defining $\mathbf{x}_t^{\text{forw},j}$ as the training-time inputs the module.

Fig. 3 shows the results. Compared with the uptrend dynamics of vanilla diffusion models in Fig. 2, we can see that the new ones are more like a slightly fluctuating horizontal line, indicating that: 1) every denoising module $p_\theta(\mathbf{x}_{t-1} \mid \mathbf{x}_t)$ has become robust to inaccurate inputs since error measure $\mathcal{D}_{\text{MMD}}(t)$ is not correlated with its location $t$ in the diffusion model; 2) By applying our regularization $\mathcal{L}^{\text{reg}}$, the diffusion model is not affected by error propagation anymore.

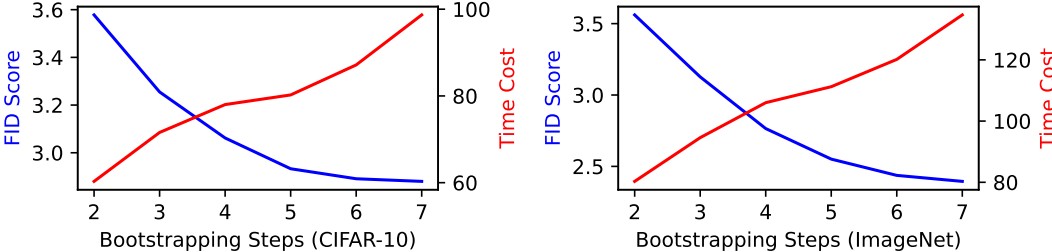

Figure 4: A trade-off study of the hyper-parameter $L$ (i.e., the number of bootstrapping steps) on two datasets. The results show that, as $L$ gets larger, the model performance limitedly increases while the time cost (measured in seconds) boundlessly grows.

| Approach | CIFAR-10 ($32 \times 32$) | CelebA ($64 \times 64$) |
|---|---|---|
| DDPM w/ Our Regularization | **2.93** | **1.22** |
| Our Model w/o Exponential Schedule $w_t$ | 3.12 | 1.35 |
| Our Model w/o Warm-start $\widetilde{\mathbf{x}}_s^{\text{back},i}$ ($L = 5$) | 3.38 | 1.45 |
| Our Model w/o Warm-start $\widetilde{\mathbf{x}}_s^{\text{back},i}$ ($L = 7$) | 3.15 | 1.27 |

Table 2: Ablation studies of the two techniques (i.e., the schedule of loss weight $w_t$ and the warm-start initialization of alternative representation $\widetilde{\mathbf{x}}_s^{\text{back},i}$) used in our approach.

### 6.3 Performances of Image Generation

The FID scores of our model and baselines on 3 datasets are demonstrated in Table 1. The results of the first row are copied from (Ning et al., 2023) and the others are obtained by our experiments. We draw two conclusions from the table: 1) Our regularization notably improves the performances of DDPM, a mainstream architecture of diffusion models, with reductions of FID scores as $18.83\%$ on CIFAR-10, $29.55\%$ on ImageNet, and $29.47\%$ on CelebA. Therefore, handling error propagation benefits in improving the generation quality of diffusion models; 2) Our model significantly outperforms ADM-IP, the baseline that reduces the exposure bias by input perturbation, on all datasets. For example, our FID score on CIFAR-10 are lower than its score by $9.84\%$. An important factor contributing to our much better performances is that, unlike the baseline, our approach doesn't impose any assumption on the distribution form of propagating errors.

### 6.4 Trade-off Study

The number of bootstrapping steps $L$ is a key hyper-parameter in our proposed regularization. Its value determination involves a trade-off between effectiveness (i.e., the quality of alternative backward variable $\widetilde{\mathbf{x}}_t^{\text{back},i}$) and run-time efficiency.

Fig. 4 shows how FID scores (i.e., model performances) and the averaged time costs of one training step change with respect to different bootstrapping steps $L$. For both CIFAR-10 and ImageNet, we can see that, as the number of steps $L$ increases, FID scores decrease and tend to converge while time costs boundlessly grow. In practice, we set $L$ as 5 for our model since this configuration leads to good enough performances and incur relatively low time costs.

### 6.5 Ablation Experiments

To make our regularization $\mathcal{L}^{\text{reg}}$ work in practice, we exponentially schedule weight $w_t, 0 \leqslant t < T$ and specially initialize alternative backward variable $\widetilde{\mathbf{x}}_s^{\text{back},i}$ via Eq. (16). Table 2 demonstrates the ablation studies of these two techniques. For the weighted schedule, we can see from the second row that model performances are notably degraded by replacing it with equal weight $w_t = 1/T$. Firstly, by adopting random initialization instead of Eq. (16), the model performances drastically decrease, implying that it's necessary to have a warm start; Secondly, the impact of initialization can be reduced by using a larger number of bootstrapping steps $L$.

ACKNOWLEDGMENTS

We thank the anonymous ICLR reviewers for their kind and constructive reviews. Yangming Li also thanks Accenture for their sponsorship and support.

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

# A  IDEAL SCENARIO: ZERO CUMULATIVE ERRORS

The error $\mathcal{E}_t^{\text{cumu}}, t \in [1, T]$ defined by KL divergence can be expanded as

$$\mathcal{E}_t^{\text{cumu}} = D_{\text{KL}}(p_\theta(\mathbf{x}_{t-1}) \parallel q(\mathbf{x}_{t-1})) = \int_{\mathbf{x}_{t-1}} p_\theta(\mathbf{x}_{t-1}) \ln \frac{p_\theta(\mathbf{x}_{t-1})}{q(\mathbf{x}_{t-1})} d\mathbf{x}_{t-1}, \qquad (18)$$

which achieves its minimum 0 exclusively at $p_\theta(\mathbf{x}_{t-1}) = q(\mathbf{x}_{t-1})$. We show that $\mathcal{E}_t^{\text{cumu}} = 0, \forall t \in [0, T]$ holds in an ideal situation (i.e., diffusion models are *perfectly optimized*).

**Proposition A.1** (Zero Cumulative Errors). *A diffusion model is perfectly optimized if* $p_\theta(\mathbf{x}_{t-1} \mid \mathbf{x}_t) = q(\mathbf{x}_{t-1} \mid \mathbf{x}_t), \forall t \in [1, T]$. *In that case, we have* $\mathcal{E}_t^{\text{cumu}} = 0, \forall t \in [1, T]$.

*Proof.* It suffices to show $p_\theta(\mathbf{x}_t) = q(\mathbf{x}_t), \forall t \in [0, T]$ for perfectly optimized diffusion models. We apply mathematical induction to prove this assertion. Initially, let's check whether this is true at $t = T$. Based on a nice property of the forward process (Ho et al., 2020):

$$q(\mathbf{x}_t \mid \mathbf{x}_0) = \mathcal{N}(\mathbf{x}_t; \sqrt{\bar{\alpha}_t} \mathbf{x}_0, (1 - \bar{\alpha}_t)\mathbf{I}),$$

we can see that term $q(\mathbf{x}_T \mid \mathbf{x}_0)$ converges to a normal Gaussian $\mathcal{N}(\mathbf{0}, \mathbf{I})$ as $T \rightarrow \infty$ because $\lim_{T \rightarrow \infty} \sqrt{\bar{\alpha}_T} = 0$. Also note that $p_\theta(\mathbf{x}_T) = p(\mathbf{x}_T) = \mathcal{N}(\mathbf{x}_T; \mathbf{0}, \mathbf{I})$, we have

$$\mathbf{q}(\mathbf{x}_T) = \int_{\mathbf{x}_0} \mathbf{q}(\mathbf{x}_T \mid \mathbf{x}_0)\mathbf{q}(\mathbf{x}_0)d\mathbf{x}_0 = \mathcal{N}(\mathbf{x}_T; \mathbf{0}, \mathbf{I}) \int_{\mathbf{x}_0} \mathbf{q}(\mathbf{x}_0)d\mathbf{x}_0 = p_\theta(\mathbf{x}_T) * 1. \qquad (19)$$

Therefore, the initial case is proved. Now, we assume the conclusion holds for $t$. Considering the precondition $p_\theta(\mathbf{x}_{t-1} \mid \mathbf{x}_t) = q(\mathbf{x}_{t-1} \mid \mathbf{x}_t)$, we have

$$\begin{aligned}
p_\theta(\mathbf{x}_{t-1}) &= \int_{\mathbf{x}_t} p_\theta(\mathbf{x}_{t-1}, \mathbf{x}_t)d\mathbf{x}_t = \int_{\mathbf{x}_t} p_\theta(\mathbf{x}_{t-1} \mid \mathbf{x}_t)p_\theta(\mathbf{x}_t)d\mathbf{x}_t \\
&= \int_{\mathbf{x}_t} q(\mathbf{x}_{t-1} \mid \mathbf{x}_t)q(\mathbf{x}_t)d\mathbf{x}_t = \int_{\mathbf{x}_t} q(\mathbf{x}_{t-1}, \mathbf{x}_t)d\mathbf{x}_t = q(\mathbf{x}_{t-1})
\end{aligned} \qquad (20)$$

which proves the case for $t - 1$. Finally, the assertion $p_\theta(\mathbf{x}_t) = q(\mathbf{x}_t)$ is true for every $t \in [0, T]$, and therefore the whole conclusion is proved.  □

In our proof to this proposition, the use of that precondition $T \rightarrow \infty$, is to let $q(x_T|x_0) \rightarrow N(x_T; 0, I)$. Considering the fact that

$$q(x_t|x_0) = N(x_t; \sqrt{\bar{\alpha}_t}x_0, \sqrt{1 - \bar{\alpha}_t}\epsilon).$$

we can achieve the goal with a much weaker assumption: $\lim_{t \rightarrow T} \bar{\alpha}_t \rightarrow 0$. Notably, this new assumption is a standard configuration in current diffusion models (e.g., DDPM (Ho et al., 2020) and SGM (Song et al., 2021b)), which lets $\mathbf{x}_T$ contain no information about $\mathbf{x}_0$.

# B  DEFINITION OF THE MODULAR ERROR

The KL divergence is not symmetric, so the modular error can have another definition as

$$\mathcal{E}_t^{\text{mod}} = \mathbb{E}_{\mathbf{x}_t \sim q(\mathbf{x}_t)}[D_{\text{KL}}(q(\mathbf{x}_{t-1} \mid \mathbf{x}_t) \parallel p_\theta(\mathbf{x}_{t-1} \mid \mathbf{x}_t))].$$

However, the error propagation happens to the learnable backward process $p_\theta$ (instead of the predefined forward process $q$). Hence, it makes more sense to define the module error as

$$\begin{aligned}
&\mathbb{E}_{\mathbf{x}_t \sim p_\theta(\mathbf{x}_t)}\Big[D_{\text{KL}}\Big(p_\theta(\mathbf{x}_{t-1} \mid \mathbf{x}_t) \parallel q(\mathbf{x}_{t-1} \mid \mathbf{x}_t)\Big)\Big] \\
&= \int p_\theta(\mathbf{x}_t)\Big(\int p_\theta(\mathbf{x}_{t-1} \mid \mathbf{x}_t) \ln \Big(\frac{p_\theta(\mathbf{x}_{t-1} \mid \mathbf{x}_t)}{q(\mathbf{x}_{t-1} \mid \mathbf{x}_t)}\Big)d\mathbf{x}_{t-1}\Big)d\mathbf{x}_t \\
&= E_{\mathbf{x}_t \sim p_\theta(\mathbf{x}_t)}\Big[E_{\mathbf{x}_t \sim p_\theta(\mathbf{x}_{t-1}|\mathbf{x}_t)}\Big[\ln \Big(\frac{p_\theta(\mathbf{x}_{t-1} \mid \mathbf{x}_t)}{q(\mathbf{x}_{t-1} \mid \mathbf{x}_t)}\Big)\Big]\Big],
\end{aligned}$$

such that expectation integrals are operated on the distributions $p_\theta(\mathbf{x}_t), p_\theta(\mathbf{x}_{t-1} \mid \mathbf{x}_t)$ of the backward process to average the distribution gap $\ln(\cdot)$. For the reversed case, the expectation operations will be mistakenly applied to the distributions $q(\mathbf{x}_t), q(\mathbf{x}_{t-1} \mid \mathbf{x}_t)$ of the forward process.

## C    PROOF TO THEOREM 3.1

The cumulative error $\mathcal{E}_t^{\text{cumu}}$ can be decomposed into two terms:

$$\mathcal{E}_t^{\text{cumu}} = \int_{\mathbf{x}_{t-1}} p_\theta(\mathbf{x}_{t-1}) \ln \frac{p_\theta(\mathbf{x}_{t-1})}{q(\mathbf{x}_{t-1})} d\mathbf{x}_{t-1} = -H_{p_\theta}(\mathbf{x}_{t-1}) + \mathbb{E}_{\mathbf{x}_{t-1}}[-\ln q(\mathbf{x}_{t-1})], \qquad (21)$$

where the second last term is the entropy of distribution $p_\theta(\mathbf{x}_t)$. We denote the last term as $\mathcal{T}_{t-1}$. According to the law of total expectation, we have

$$\mathcal{T}_{t-1} = \mathbb{E}_{\mathbf{x}_t \sim p_\theta(\mathbf{x}_t)}[\mathbb{E}_{\mathbf{x}_{t-1} \sim p_\theta(\mathbf{x}_{t-1}|\mathbf{x}_t)}[-\ln q(\mathbf{x}_{t-1})]]. \qquad (22)$$

Note that $q(\mathbf{x}_{t-1})q(\mathbf{x}_t \mid \mathbf{x}_{t-1}) = q(\mathbf{x}_{t-1}, \mathbf{x}_t) = q(\mathbf{x}_t)q(\mathbf{x}_{t-1} \mid \mathbf{x}_t)$, we have

$$\mathcal{T}_{t-1} = \mathbb{E}_{\mathbf{x}_t \sim p_\theta(\mathbf{x}_t)}\left[\mathbb{E}_{\mathbf{x}_{t-1} \sim p_\theta(\mathbf{x}_{t-1}|\mathbf{x}_t)}\left[-\ln q(\mathbf{x}_t) - \ln \frac{q(\mathbf{x}_{t-1} \mid \mathbf{x}_t)}{q(\mathbf{x}_t \mid \mathbf{x}_{t-1})}\right]\right]$$

$$= \mathbb{E}_{\mathbf{x}_t \sim p_\theta(\mathbf{x}_t)}\left[-\ln q(\mathbf{x}_t) + \mathbb{E}_{\mathbf{x}_{t-1} \sim p_\theta(\mathbf{x}_{t-1}|\mathbf{x}_t)}\left[-\ln \frac{q(\mathbf{x}_{t-1} \mid \mathbf{x}_t)}{q(\mathbf{x}_t \mid \mathbf{x}_{t-1})}\right]\right]. \qquad (23)$$

$$= \mathcal{T}_t + \mathbb{E}_{\mathbf{x}_t \sim p_\theta(\mathbf{x}_t)}\left[\mathbb{E}_{\mathbf{x}_{t-1} \sim p_\theta(\mathbf{x}_{t-1}|\mathbf{x}_t)}\left[\ln \frac{q(\mathbf{x}_t \mid \mathbf{x}_{t-1})}{q(\mathbf{x}_{t-1} \mid \mathbf{x}_t)}\right]\right]$$

Now, we focus on the iterated expectations $\mathbb{E}_{\mathbf{x}_{t+1}}[\mathbb{E}_{\mathbf{x}_t}[\cdot]]$ of the above equality. Based on the definition of KL divergence, it's obvious that

$$\mathbb{E}_{\mathbf{x}_t \sim p_\theta(\mathbf{x}_t)}\left[\mathbb{E}_{\mathbf{x}_{t-1} \sim p_\theta(\mathbf{x}_{t-1}|\mathbf{x}_t)}\left[\ln \left(\frac{p_\theta(\mathbf{x}_{t-1} \mid \mathbf{x}_t)}{q(\mathbf{x}_{t-1} \mid \mathbf{x}_t)} \frac{q(\mathbf{x}_t \mid \mathbf{x}_{t-1})}{p_\theta(\mathbf{x}_{t-1} \mid \mathbf{x}_t)}\right)\right]\right]$$

$$= \mathbb{E}_{\mathbf{x}_t \sim p_\theta(\mathbf{x}_t)}\left[\mathbb{E}_{\mathbf{x}_{t-1} \sim p_\theta(\mathbf{x}_{t-1}|\mathbf{x}_t)}\left[\ln \frac{p_\theta(\mathbf{x}_{t-1} \mid \mathbf{x}_t)}{q(\mathbf{x}_{t-1} \mid \mathbf{x}_t)}\right] + \mathbb{E}_{\mathbf{x}_{t-1}}\left[\ln \frac{q(\mathbf{x}_t \mid \mathbf{x}_{t-1})}{p_\theta(\mathbf{x}_{t-1} \mid \mathbf{x}_t)}\right]\right]. \qquad (24)$$

$$= \mathbb{E}_{\mathbf{x}_t \sim p_\theta(\mathbf{x}_t)}[D_{\text{KL}}(\cdot)] + \mathbb{E}_{\mathbf{x}_t \sim p_\theta(\mathbf{x}_t)}\left[\mathbb{E}_{\mathbf{x}_{t-1} \sim p_\theta(\mathbf{x}_{t-1}|\mathbf{x}_t)}\left[\ln \frac{q(\mathbf{x}_t \mid \mathbf{x}_{t-1})}{p_\theta(\mathbf{x}_{t-1} \mid \mathbf{x}_t)}\right]\right]$$

We denote term $\mathbb{E}_{\mathbf{x}_t}[\mathbb{E}_{\mathbf{x}_{t-1} \sim p_\theta(\mathbf{x}_{t-1}|\mathbf{x}_t)}[\cdot]]$ here as $\mathcal{I}_t$ and prove that it is a constant. Since the distribution $q(\mathbf{x}_t \mid \mathbf{x}_{t-1})$ is a predefined multivariate Gaussian, we have

$$q(\mathbf{x}_t \mid \mathbf{x}_{t-1}) = (2\pi\beta_t)^{-\frac{K}{2}} \exp\left(-\frac{\|\mathbf{x}_t - \sqrt{1-\beta_t}\mathbf{x}_{t-1}\|^2}{2\beta_t}\right)$$

$$= (1-\beta_t)^{-\frac{K}{2}}(2\pi\beta_t/(1-\beta_t))^{-\frac{K}{2}} \exp\left(-\frac{\|\mathbf{x}_{t-1} - (\mathbf{x}_t/\sqrt{1-\beta_t})\|^2}{2\beta_t/(1-\beta_t)}\right). \qquad (25)$$

$$= (1-\beta_t)^{-\frac{K}{2}} \mathcal{N}(\mathbf{x}_{t-1}; \mathbf{x}_t/\sqrt{1-\beta_t}, \beta_t/(1-\beta_t)\mathbf{I})$$

With this result and the definition of learnable backward probability $p_\theta(\mathbf{x}_{t-1} \mid \mathbf{x}_t)$, we can convert the constant $\mathcal{I}_t$ into the following form:

$$\mathcal{I}_t = \mathbb{E}_{\mathbf{x}_t}\left[-\mathbb{E}_{\mathbf{x}_{t-1}}\left[\ln \frac{\mathcal{N}(\mathbf{x}_{t-1}; \boldsymbol{\mu}_\theta(\mathbf{x}_t, t), \sigma_t \mathbf{I})}{\mathcal{N}(\mathbf{x}_{t-1}; \mathbf{x}_t/\sqrt{1-\beta_t}, \beta_t/(1-\beta_t)\mathbf{I})}\right]\right] - \frac{K}{2}\ln(1-\beta_t). \qquad (26)$$

Note that term $\mathbb{E}_{\mathbf{x}_{t-1}}[\cdot]$ essentially represents the KL divergence between two Gaussian distributions, which is said to have a closed-form solution (Zhang et al., 2021):

$$D_{\text{KL}}\left(\mathcal{N}\left(\mathbf{x}_{t-1}; \boldsymbol{\mu}_\theta(\mathbf{x}_t, t), \sigma_t \mathbf{I}\right) \| \mathcal{N}\left(\mathbf{x}_{t-1}; \frac{\mathbf{x}_t}{\sqrt{1-\beta_t}}, \frac{\beta_t}{1-\beta_t}\mathbf{I}\right)\right)$$

$$= \frac{1}{2}\left(K \ln \frac{\beta_t}{(1-\beta_t)\sigma_t} - K + \frac{(1-\beta_t)\sigma_t}{\beta_t}K + \frac{1-\beta_t}{\beta_t}\left\|\frac{\mathbf{x}_t}{\sqrt{1-\beta_t}} - \boldsymbol{\mu}_\theta(\cdot)\right\|^2\right).$$

Considering this result, the fact that variance $\sigma_t$ is primarily set as $\beta_t$ in DDPM (Ho et al., 2020), and the definition of mean $\boldsymbol{\mu}_\theta$, we can simplify term $\mathcal{I}_t$ as

$$\mathcal{I}_t = \frac{1-\beta_t}{2\beta_t}\mathbb{E}_{\mathbf{x}_t \sim p_\theta(\mathbf{x}_t)}\left[\left\|\frac{\mathbf{x}_t}{\sqrt{1-\beta_t}} - \boldsymbol{\mu}_\theta(\mathbf{x}_t, t)\right\|^2\right] - \frac{\beta_t}{2}$$

$$= \frac{\beta_t}{2K(1-\bar{\alpha}_t)}\mathbb{E}_{\mathbf{x}_t \sim p_\theta(\mathbf{x}_t)}[\|\boldsymbol{\epsilon}_\theta(\mathbf{x}_t, t)\|^2] - \frac{\beta_t}{2} \qquad (27)$$

Considering our assumption that neural network $\epsilon_\theta$ behaves as a standard Gaussian irrespective of the input distribution, we can interpret term $\mathbb{E}_{\mathbf{x}_t \sim p_\theta(\mathbf{x}_t)}[\cdot]$ as the total variance of all dimensions of a multivariate normal distribution. Therefore, its value is $K \times 1$, the sum of $K$ unit variances. With this result, term $\mathcal{I}_t$ can be reduced as

$$\mathcal{I}_t = \frac{\beta_t}{2}\Big(\frac{1}{1-\bar{\alpha}_t} - 1\Big) = \frac{\beta_t \bar{\alpha}_t}{1-\bar{\alpha}_t} > 0. \tag{28}$$

Combining this inequality with Eq. (23) and Eq. (24), we have

$$\mathcal{T}_{t-1} = \mathcal{T}_t + \mathbb{E}_{\mathbf{x}_t \sim p_\theta(\mathbf{x}_t)}[D_{\mathrm{KL}}(p_\theta(\mathbf{x}_{t-1} \mid \mathbf{x}_t) \,||\, q(\mathbf{x}_{t-1} \mid \mathbf{x}_t))] + \mathcal{I}_t = \mathcal{T}_t + \mathcal{E}_t^{\mathrm{mod}} + \mathcal{I}_t, \tag{29}$$

and its derivation $\mathcal{T}_{t-1} \geqslant \mathcal{T}_t + \mathcal{E}_t^{\mathrm{mod}}$ are both true.

Considering the assumption that the entropy of distribution $p_\theta(\mathbf{x}_t)$ decreases during the backward process (i.e., $H_{p_\theta}(\mathbf{x}_{t-1}) < H_{p_\theta}(\mathbf{x}_t), 1 \leqslant t \leqslant T$), we have

$$\mathcal{E}_t^{\mathrm{cumu}} \geqslant -H_{p_\theta}(\mathbf{x}_t) + \mathcal{T}_t + \mathcal{E}_t^{\mathrm{mod}} = \mathcal{E}_{t+1}^{\mathrm{cumu}} + \mathcal{E}_t^{\mathrm{mod}}, \tag{30}$$

which is exactly the propagation equation.

## D  WEAKENED ASSUMPTION TO THEOREM 3.1

Since neural network $\epsilon_\theta(\mathbf{x}_t, t)$ is designed to fit Gaussian noise $\epsilon$ in the loss function $\mathcal{L}^{\mathrm{nll}}$, , we previously supposed that its output distribution follows a standard Gaussian. In our proof to Theorem 3.1, that assumption is applied to indicate that $\mathbb{E}[|\epsilon_\theta(\mathbf{x}_t, t)|^2] = K$. ($K$ is the vector dimension). However, considering Eq. (27) and Eq. (28) in the proof, our theorem still holds for $\mathbb{E}[|\epsilon_\theta(\mathbf{x}_t, t)|^2] \geqslant K$. To derive a new assumption, we first set a term

$$r_t = \mathbb{E}[|\epsilon - \epsilon_\theta(\sqrt{\bar{\alpha}_t}\mathbf{x}_0 + \sqrt{1-\bar{\alpha}_t}\epsilon, t)|^2], \tag{31}$$

which indicates the prediction error of neural network $\epsilon_\theta(\mathbf{x}_t, t)$. The term will vanish to 0 if the neural network is fully accurate in backward denoising. According to the Triangle Inequality, we then have

$$r_t \geqslant \mathbb{E}[|\epsilon|^2 - |\epsilon_\theta(\cdot)|^2] = \mathbb{E}[|\epsilon|^2] - \mathbb{E}[|\epsilon_\theta(\cdot)|^2]. \tag{32}$$

Since the second moment of Gaussian distribution $\mathbb{E}[|\epsilon|^2]$ is $K$, we further have

$$(1/K)\mathbb{E}[|\epsilon_\theta(\sqrt{\bar{\alpha}_t}\mathbf{x}_0 + \sqrt{1-\bar{\alpha}_t}\epsilon, t)|^2] \geqslant 1 - (r_t/K). \tag{33}$$

Finally, consider the fact (Ho et al., 2020) that $\mathbf{x}_t$ can be reparameterized as $\sqrt{\bar{\alpha}_t}\mathbf{x}_0 + \sqrt{1-\bar{\alpha}_t}\epsilon, \epsilon \sim \mathcal{N}(\mathbf{0}, \mathbf{I})$ and let the prediction error $r_t$ vanish, the above equation motivates us to make the following new assumption:

$$(1/K)\mathbb{E}[|\epsilon_\theta(\mathbf{x}_t, t)|^2] \geqslant 1, \tag{34}$$

which is much weaker than the previous assumption but still makes our Theorem 3.1 hold.

## E  PROOF TO PROPOSITION 4.1

Theorem 3 of Wang & Tay (2022) indicates that the following equation is true:

$$-\ln(1 - \frac{1}{4}\mathrm{MMD}^2(\mathcal{C}_\infty(\Omega), \mathbb{P}, \mathbb{Q})) \leqslant D_{\mathrm{KL}}(\mathbb{P} \,||\, \mathbb{Q}) \leqslant \ln(\mathrm{MMD}^2(\mathcal{C}(\Omega, \mathcal{Q}), \mathbb{P}, \mathbb{Q}) + 1), \tag{35}$$

if with the Assumption 1 of Wang & Tay (2022) and $\frac{d\mathbb{P}}{d\mathbb{Q}}$ is continuous on $\Omega$.

By setting $\Omega = \mathbb{R}^K, \mathbb{P}(\cdot) = \int p_\theta(\mathbf{x}_{t-1})d\mathbf{x}_{t-1}, \mathbb{Q}(\cdot) = \int q(\mathbf{x}_{t-1})d\mathbf{x}_{t-1}$ and supposing that $p_\theta(\mathbf{x}_{t-1})$, $q(\mathbf{x}_{t-1})$ are smooth and non-zero everywhere, the two conditions of Eq. (35) are both met. Furthermore, Wang & Tay (2022) stated that $\mathcal{C}_\infty(\Omega)$ is a subset of $\mathcal{C}(\Omega, \mathcal{Q})$. Considering the definitions of the MMD and cumulative errors, we can turn Eq. (35) into the following equation:

$$-\ln(1 - \frac{1}{4}\mathcal{D}_t^{\mathrm{cumu}}) \leqslant \mathcal{E}_t^{\mathrm{cumu}} \leqslant \ln(\mathcal{D}_t^{\mathrm{cumu}} + 1). \tag{36}$$

Since $\ln(x+1) \leqslant x, -\ln(1 - \frac{1}{4}x) \geqslant \frac{1}{4}$, we can get

$$
\begin{cases}
\mathcal{E}_t^{\text{cumu}} \leqslant \ln(\mathcal{D}_t^{\text{cumu}} + 1) \leqslant \mathcal{D}_t^{\text{cumu}} \\
\mathcal{E}_t^{\text{cumu}} \geqslant -\ln(1 - \frac{1}{4}\mathcal{D}_t^{\text{cumu}}) \geqslant \frac{1}{4}\mathcal{D}_t^{\text{cumu}}
\end{cases},
\tag{37}
$$

which are exactly our expected conclusion.

Strictly speaking, we also have to verify that $D_t^{cumu} < 4$ such that the inequality $-\ln(1 - \frac{1}{4}x) \geqslant \frac{1}{4}$ is properly applied. Recall that the definition of $\mathcal{D}_t^{\text{cumu}}$ as

$$
|\mathbb{E}_{p_\theta(\mathbf{x}_{t-1})}[\phi(\mathbf{x}_{t-1})] - \mathbb{E}_{q(\mathbf{x}_{t-1})}[\phi(\mathbf{x}_{t-1})]|^2,
$$

where $|\phi| = \sup_{\mathbf{x}} |\phi(\mathbf{x})| < 1$. According to the Triangle Inequality, we have,

$$
\mathcal{D}_t^{\text{cumu}} \leqslant (|\mathbb{E}_{p_\theta(\mathbf{x}_{t-1})}[\phi(\mathbf{x}_{t-1})]| + |\mathbb{E}_{q(\mathbf{x}_{t-1})}[\phi(\mathbf{x}_{t-1})]|)^2
\tag{38}
$$

Since $|x|$ is a convex function, we can apply Jensen's inequality to the above equation:

$$
\begin{aligned}
\mathcal{D}_t^{\text{cumu}} &\leqslant (E_{p_\theta(x_{t-1})}[|\phi(x_{t-1})|] + E_{q(x_{t-1})}[|\phi(x_{t-1})|])^2 \\
&< (E_{p_\theta(x_{t-1})}[1] + E_{q(x_{t-1})}[1])^2 = (1+1)^2 = 4,
\end{aligned}
\tag{39}
$$

showing that our claim holds.

## F  TRAINING ALGORITHM

Compared with common practices (Ho et al., 2020; Song et al., 2021a), we additionally regularize the optimization of DMs with *cumulative errors*. The details are in Algo. 1.

---

**Algorithm 1:** Optimization with Our Proposed Regularization

**Input:** Batch size $B$, number of backward iterations $T$, sample range $L \ll T$.

**while** *the model is not converged* **do**

    Sample a batch of samples from the training set $\mathcal{S}_0 = \{\mathbf{x}_0^i \mid \mathbf{x}_0^i \sim q(\mathbf{x}_0^i), 1 \leqslant i \leqslant T\}$.

    Sample a time point for vanilla training $t \in \mathcal{U}\{1, T\}$.

    Estimate training loss $\mathcal{L}_t^{\text{nll}}$ for every real sample $\mathbf{x}_0^i \in \mathcal{S}_0$.

    Sample a time point for regularization $s \in \mathcal{U}\{\min(t+L, T), t+1\}$.

    Sample a new batch of samples from the training set $\mathcal{S}_0' = \{\mathbf{x}_0^j \mid \mathbf{x}_0^j \sim q(\mathbf{x}_0^j), 1 \leqslant j \leqslant T\}$.

    Sample forward variables $\mathcal{S}_s^{\text{forw}} = \{\mathbf{x}_s^{\text{forw},j} \mid \mathbf{x}_s^{\text{forw},j} \sim q(\mathbf{x}_s^{\text{forw},j} \mid \mathbf{x}_0^j), \mathbf{x}_0^j \in \mathcal{S}_0'\}$ at step $s$.

    Sample forward variables $\mathcal{S}_t^{\text{forw}} = \{\mathbf{x}_t^{\text{forw},i} \mid \mathbf{x}_t^{\text{forw},i} \sim q(\mathbf{x}_t^{\text{forw},i} \mid \mathbf{x}_0^i), \mathbf{x}_0^i \in \mathcal{S}_0\}$ at step $t$.

    Sample alternative backward variables at time step $s$ as

    $\mathcal{S}_s^{\text{back}} = \{\tilde{\mathbf{x}}_s^{\text{back},i} \mid \tilde{\mathbf{x}}_s^{\text{back},i} \sim p_\theta(\tilde{\mathbf{x}}_s^{\text{back},i} \mid \mathbf{x}_t^{\text{forw},i}), \mathbf{x}_0^i \in \mathcal{S}_t^{\text{forw}}\}$.

    Estimate *cumulative error* $\mathcal{L}_s^{\text{reg}}$ based on $\mathcal{S}_s^{\text{forw}}$ and $\mathcal{S}_s^{\text{back}}$.

    Update the model parameter $\theta$ with gradient $\nabla_\theta(\lambda^{\text{nll}}\mathcal{L}_t^{\text{nll}} + \lambda^{\text{reg}}w_s\mathcal{L}_s^{\text{reg}})$.

---

## G  RELATED WORK

**Exposure bias.**   The topic of our paper is closely related to a problem called exposure bias that occurs to some sequence models (Ranzato et al., 2016; Zhang et al., 2019), which means that a model only exposed to ground truth inputs might not be robust to errors during evaluation. (Ning et al., 2023; Li et al., 2023) studied this problem for diffusion models due to their sequential structure. However, they lack a solid explanation of why the models are not robust to exposure bias, which is very important because many sequence models (e.g., CRF) is free from this problem. Our analysis in Sec. 3 actually answers that question with empirical evidence and theoretical analysis. More importantly, we argue that ADM-IP (Ning et al., 2023), which adds an extra Gaussian noise into the model input, is not an ideal solution, since this simple perturbation certainly can not simulate the complex errors at test time. We have also treated their approach as a baseline and shown that our regularization significantly outperforms it in experiments (Section 6).

**Consistency regularizations.** Some papers (Lai et al., 2022; Daras et al., 2023; Song et al., 2023; Lai et al., 2023), which aim to improve the estimation of score functions. One big difference between these papers and our work is that they assert without proof that diffusion models are affected by error propagation, while we have empirically and theoretically verified whether this phenomenon happens. Notably, this assertion is not trivial because many sequential models (e.g., CRF and HMM) are free from error propagation. Besides, these papers proposed regularisation methods in light of some expected consistencies. While these methods are similar to our regularisation in name, they were actually to improve the score estimation at every time step (i.e., the prediction accuracy of every component in a sequential model), which differs much from our approach that aims to improve the robustness of score functions to input errors. The key to solving error propagation is to make score functions insensitive to the input *cumulative errors* (rather than pursuing higher accuracy) (Motter & Lai, 2002; Crucitti et al., 2004), because it's very hard to have perfect score functions with limited training data. Therefore, these methods are orthogonal to us in reducing the effect of error propagation and do not constitute ideal solutions to the problem.

**Convergence guarantees.** Another group of seemingly related papers (Lee et al., 2022; Chen et al., 2023b;a) aim to derive convergence guarantees based on the assumption of bounded score estimation errors. Specifically speaking, these papers analysed how generated samples converge (in distribution) to real data with respect to increasing discretization levels. However, studies (Motter & Lai, 2002; Crucitti et al., 2004) on error propagation generally focus on analysing the error dynamics of a chain model over time (i.e., how input errors to the score functions of decreasing time steps evolve). Therefore, these papers are of a very different research theme from our paper. More notably, these papers either assumed bounded score estimation errors or just ignored them, which are actually not appropriate to analyse error propagation for diffusion models. There are two reasons. Firstly, the error dynamics shown in our paper (Fig. 2) have exponentially-like increasing trends, implying that the score functions close to time step 0 are of very poor estimation. Secondly, if all components of a chain model are very accurate (i.e., bounded estimation errors), then the effect of error propagation will be insignificant regardless of the values of amplification factor. Therefore, it's better to set the estimation errors as uncertain variables for studying error propagation

