# OpenReview forum: "On Error Propagation of Diffusion Models"
_ICLR.cc/2024/Conference — ICLR 2024 poster_

### Official Review · Reviewer_ebh3 · 2023-10-28

**Soundness:** 3 good
**Presentation:** 4 excellent
**Contribution:** 3 good
**Rating:** 8
**Confidence:** 4

**Summary:**

This paper presents a theoretical framework to quantify per iteration error in variational inference within the backward pass of difussion models.
The authors introduce metrics to theoretically quantify these errors and convincingly demonstrate their increase with iteration number.
Consequently, the model's performance deteriorates as the number of iterations increases.
To mitigate this issue, the authors propose to add the error as a regularization term upon the original loss function.
Empirical experiments are conducted to validate the effectiveness of this proposed methodology.

**Strengths:**

- The paper is very well written and I really enjoys reading this paper.

- The concept of error quantification presented by the authors is both innovative and well-conceived.

**Weaknesses:**

- Modular Error Definition: The modular error is defined as the expected KL divergence from $p_{\theta}$ to $q$. However, the KL divergence is not symmetric. Could the authors please explain why the modular error is defined in such a way? Why not using the reversed KL divergence as in the original loss function?


- Assumptions in Theorem 3.1: The assumptions regarding the output distribution of the neural network (NN) following a standard Gaussian distribution and the entropy reduction with iteration number appear quite strong. Can the authors justify these two assumptions empirically?


- Technical Issues in the Proof:

    - In the proof of Proposition A.1 in Appendix A, equation (19) holds only in the limit as $T\to\infty$. The current proof falls short of demonstrating Proposition A.1 for finite values of $T$.

    - Several typographical errors in equation (24) in Appendix B raise concerns about the validity of the proof. Specifically, the $p_{\theta}$ term in both the numerator and denominator should be consistent and denoted as $p_{\theta}(x_{t-1}|x_{t})$ in order for equation (29) to hold. Additionally, the first term in the second line of (24) appears to be missing a logarithmic notation, and the third line of (24) appears inconsistent with the current notation.

- Hyper-Parameter Selection: The authors should provide further clarification on the process for selecting hyper-parameters. Notably, the weight assigned to the regularization term is a critical factor that requires elucidation.
How to select the hyper-parameters? One such important factor is the weight of the regularization term, could the authors please clarify?

- Minor Issues:

    - The notation for conditional expectation should be corrected throughout the paper to $E_{X_{t}|X_{t+1}}$.

    - In Section 3.2, second paragraph, "get ride of" should be corrected to "get rid of".

    - The proof for (14), establishing the boundedness of the KL divergence by the MMD, rests on the assumption that $\log(1-x/4)$ is well-defined, which only holds when $x<4$. The authors should address and clarify this point.

**Questions:**

Please refer to the questions in the previous section.

---

> ### Author Response · Authors · 2023-11-14
> **Part-1 of Our Response**
>
> Dear Reviewer ebh3,
>
> We thank you for your kind, constructive, and comprehensive feedback. In the following, we have answered all your concerns in a point-by-point manner.
>
> ### Comment-1: Modular Error Definition … the KL divergence is not symmetric. Could the authors please explain why the modular error is defined in such a way? …
>
> Answer-1: Thanks for raising this insightful point. We did consider the asymmetry of KL divergence. The key is that **the error propagation happens to the learnable backward process $p_{\theta}$** (instead of the predefined forward process $q$). Therefore, it makes more sense to define the module error as
>
> $E_{x_t \sim p_{\theta}(x_t)}[ D_{KL} ( p_{\theta}(x_{t-1} | x_t) \mid\mid q(x_{t-1} | x_t) ] = \int p_{\theta}(x_t) ( \int p_{\theta}(x_{t-1} | x_t) \ln (\cdot) dx_{t-1}) dx_t = E_{x_t \sim p_{\theta}(x_t)}[ E_{x_t \sim p_{\theta}( x_{t-1} | x_t)}[ \ln(\cdot) ]  ] $,
>
> such that **expectation integrals are operated on the distributions $p_{\theta}(x_t), p_{\theta}(x_{t-1} | x_t)$ of the backward process** to average the distribution gap $\ln(\cdot)$. For your mentioned reversed case, the expectation operations will be mistakenly applied to the distributions $q(x_t), q(x_{t-1} | x_t)$ of the forward process.
>
> ### Comment-2: Assumptions in Theorem 3.1 … Can the authors justify these two assumptions empirically?
>
> Answer-2: Thanks for raising this point. For our assumption on $\epsilon_{\theta}(x_t, t)$, **we first weaken it into a new assumption from a theoretical angle and then perform experiments to empirically verify its validity**.
>
> Since neural network $\epsilon_{\theta}(x_t, t)$ is designed to fit Gaussian noise $\epsilon$ in the loss function $L^{nll}$, we previously supposed that the output distribution of $\epsilon_{\theta}(x_t, t)$ follows a standard Gaussian. In our proof to Theorem 3.1, **that assumption is applied to indicate that $E[\| \epsilon_{\theta}(x_t, t) \|^2] = K$** ($K$ is the vector dimension). However, considering Eq. (27) and Eq. (28) in the proof, **our theorem still holds for $E[\| \epsilon_{\theta}(x_t, t) \|^2] \ge K$**.
>
> To derive a new assumption, we first set a term
>
> $r_t = E[ \| \epsilon - \epsilon_{\theta}(\sqrt{\bar{\alpha}_t} x_0 + \sqrt{1 - \bar{\alpha}_t} \epsilon, t) \|^2 ]$,
>
> which indicates the prediction error of neural network $\epsilon_{\theta}(x_t, t)$. The term will vanish to $0$ if $\epsilon_{\theta}(x_t, t)$ is fully accurate in backward denoising. According to the Triangle Inequality, we then have
>
> $r_t \ge E[ \| \epsilon \|^2 - \| \epsilon_{\theta}(\cdot) \|^2  ] = E [\| \epsilon \|^2] - E[ \| \epsilon_{\theta}(\cdot) \|^2 ] $.
>
> Since the second moment of Gaussian distribution $E[\| \epsilon \|^2]$ is $K$, we further have
>
> $(1 / K) E[ \| \epsilon_{\theta}(\sqrt{\bar{\alpha}_t} x_0 + \sqrt{1 - \bar{\alpha}_t} \epsilon, t) \|^2 ] \ge 1 - (r_t / K) $.
>
> Finally, consider the fact [1] that $x_t$ can be reparameterized as $\sqrt{\bar{\alpha}_t} x_0 + \sqrt{1 - \bar{\alpha}_t} \epsilon, \epsilon \sim N(0, I) $ and let the prediction error $r_t$ vanish, **the above equation motivates us to make the following new assumption**:
>
> $(1 / K) E[ \| \epsilon_{\theta}(x_t, t) \|^2 ] \ge 1$,
>
> **which is much weaker than the previous assumption but still makes our Theorem 3.1 hold**.
>
> We conduct experiments on 2 datasets to verify our new assumption. For every dataset, we sample $10000$ trajectories $x_{0:T}$ from a well-trained diffusion model. In this process, we collect $10000$ outputs from neural network $\epsilon_{\theta}(x_t, t)$ at every step $t$ and use them to estimate $(1 / K) E[ \| \epsilon_{\theta}(x_t, t) \|^2 ]$. Below are the experiment results.
>
> | Dataset  | t=1000 | t=800 | t=600 | t=400 | t=200 | t=1   |
> |----------|--------|-------|-------|-------|-------|-------|
> | CIFAR-10 | 1.17   | 1.19  | 1.15  | 1.11  | 1.08  | 1.05  |
> | CelebA   | 1.25   | 1.21  | 1.23  | 1.17  | 1.11  | 1.09  |
>
> From this table, we can see that **the estimated value of $(1 / K) E[ \| \epsilon_{\theta}(x_t, t) \|^2 ]$ is consistently larger than $1$ at every backward iteration for both datasets**. Therefore, our new assumption $(1 / K) E[ \| \epsilon_{\theta}(x_t, t) \|^2 ] \ge 1$ has firm empirical support.
>
> Because it is computationally infeasible to estimate the probabilistic density $p_{\theta}(x_t)$ for discrete-time diffusion models (e.g., DDPM [1]), we find it hard to empirically verify our assumption on the entropy $H_{p_{\theta}(x_t)}$. However, we believe that assumption is very intuitive because the uncertainty of variable $x_t$ (i.e., entropy) should decrease as the denoising process progresses. For example, if we are sampling from a diffusion model and find that $x_{100}$ seems like a dog, it’s less likely that the final outcome $x_0$ (that is denoised from $x_{100}$) will be a cat.
>
> We will add the above new assumption, derivation, experiment, and discussion to the revised version.

---

> ### Author Response · Authors · 2023-11-14
> **Part-2 of Our Response**
>
> ### Comment-3: In the proof of Proposition A.1 in Appendix A … The current proof falls short … for finite values of T.
>
> Answer-3: Thanks for pointing out this. We address your concern by **introducing a weaker precondition, which still makes our proposition hold and is also supported by current works [1,2]**.
>
> In our proof to Proposition A.1, the use of that precondition $T \rightarrow \infty$ is to let $q(x_T | x_0) \rightarrow N(x_T; 0, I)$. Considering the fact that
>
> $q(x_t | x_0) = N(x_t; \sqrt{\bar{\alpha}_t} x_0, \sqrt{1 - \bar{\alpha}_t} \epsilon)$,
>
> we can achieve the goal with a much weaker assumption: $\lim_{t \rightarrow T} \bar{\alpha_t} \rightarrow 0$. Notably, **this new assumption is a standard configuration in current diffusion models** (e.g., DDPM [1] and SGM [2]), which lets $x_T$ contain no information about $x_0$.
>
> ### Comment-4: Several typographical errors in equation (24) in Appendix B … both the numerator and denominator should be consistent …  the first term in the second line of (24) … the third line of (24) appears inconsistent  …
>
> Answer-4: Thanks for pointing out this. The correct forms of the first two lines of our Eq. (24) are
>
> $ E_{x_t \sim p_{\theta}(x_{t})} [E_{x_{t-1} \sim  p_{\theta}(x_{t-1} \mid x_{t}) } [\ln ( \frac{p_{\theta}(x_{t-1} \mid x_{t})} {q(x_{t-1} \mid x_{t})} \frac{q(x_{t} \mid x_{t-1})} {p_{\theta}(x_{t-1} \mid x_{t})} ) ] ] = E_{x_t \sim p_{\theta}(x_{t})} [E_{x_{t-1} \sim  p_{\theta}(x_{t-1} \mid x_{t}) } [\ln  \frac{p_{\theta}(x_{t-1} \mid x_{t})} {q(x_{t-1} \mid x_{t})}  ] ] + E_{x_t } [E_{x_{t-1}} [ \ln \frac{q(x_{t} \mid x_{t-1})} {p_{\theta}(x_{t-1} \mid x_{t})} ] ]$.
>
> We also fix the last line of Eq. (24) as
>
> $E_{x_t \sim p_{\theta}(x_{t})} [ D_{KL} (p_{\theta}(x_{t-1} \mid x_{t}) || q(x_{t-1} \mid x_{t})) ] + E_{x_t \sim p_{\theta}(x_{t})} [E_{x_{t-1} \sim  p_{\theta}(x_{t-1} \mid x_{t}) } [ \ln \frac{q(x_{t} \mid x_{t-1})} {p_{\theta}(x_{t-1} \mid x_{t})} ] ]$.
>
> We will add the above corrections to the revised version.
>
> ### Comment-5: … clarification on the process for selecting hyper-parameters … the weight assigned to the regularization term …
>
> Answer-5: We applied the grid search to hyper-parameter selection. For the weights of regularization terms, we respectively construct candidate sets $[ 0.2, 0.4, 0.6, 0.8 ]$ and $[ 1 \times 10^{-3}, 3 \times 10^{-3}, 6 \times 10^{-3}, 9 \times 10^{-3} ]$ for $\lambda^{reg}$ and $\rho$. As mentioned in Sec. 6.1 of our paper, the experiments turned out that this combination $\lambda^{reg} = 0.2, \rho = 3 * 10^{-3}$ performed the best.
>
> ### Comment-6: Minor Issues: … The notation for conditional expectation … "get ride of" should be corrected to "get rid of" …  rests on the assumption that log(1 - x/4) is well-defined, which only holds when x < 4 …
>
> Answer-6: Thanks for raising these points. We will accept your suggestions about the notation and the typo in the revised version.
>
> For your comment “x < 4”, we prove that $D_{t}^{cumu} < 4$ such that $x = D_{t}^{cumu} < 4$. Recall that the definition of $D_{t}^{cumu}$ is as
>
> $|E_{p_{\theta}(x_{t-1})}[\phi(x_{t-1})] - E_{q(x_{t-1})}[\phi(x_{t-1})]|^2$,
>
> where $|\phi| = \sup_{x} |\phi(x)| < 1$. According to the Triangle Inequality, we have
>
> $D_{t}^{cumu} \le (|E_{p_{\theta}(x_{t-1})}[\phi(x_{t-1})]|+ |E_{q(x_{t-1})}[\phi(x_{t-1})]|)^2$,
>
> Since $|x|$ is a convex function, we can apply Jensen's inequality to the above equation:
>
> $D_{t}^{cumu} \le (E_{p_{\theta}(x_{t-1})}[|\phi(x_{t-1})|]+ E_{q(x_{t-1})}[|\phi(x_{t-1})|])^2 < (E_{p_{\theta}(x_{t-1})}[1]+ E_{q(x_{t-1})}[1])^2 = (1 + 1)^2 = 4$,
>
> showing that our claim holds.
>
> Besides the above proof, the experiments in Fig. 2 of our paper also empirically confirmed that $D_{t}^{cumu} < 4$.
>
> ## References
>
> [1] Ho et al., Denoising Diffusion Probabilistic Models, NeurlPS-2020.
>
> [2] Song et al, Score-Based Generative Modeling through Stochastic Differential Equations, ICLR-2021.

---

> ### Comment · Reviewer_ebh3 · 2023-11-20
>
> The authors have addressed my concerns and I will keep my rating as before.

---

> > ### Author Response · Authors · 2023-11-21
> >
> > Thank you, and thanks again for your constructive review!

---

### Official Review · Reviewer_hknr · 2023-10-31

**Soundness:** 3 good
**Presentation:** 3 good
**Contribution:** 2 fair
**Rating:** 8
**Confidence:** 2

**Summary:**

This paper analyses the error propagation/accumulation in DMs across iterations. The authors prove and empirically verify that the error in DMs cumulatively increases. To minimize this cumulative error as regularization, the authors prove tractable estimates which tightly bound this error, which is then used as a proxy. The authors empirically show the proposed method reduces the cumulative error, and increases generation quality across multiple datasets.

**Strengths:**

1. The theoretical framework and the bounds on error propagation through DMs are useful for analyzing robustness of DMs.
1. The proposed method results in strong significant improvements across a range of datasets.
1. The proposed method successfully decreases cumulative error in DMs

**Weaknesses:**

1. The proposed method requires significant compute overhead, so gains need to be weighed against this increase in compute.

**Questions:**

1. Contemporaneous work[1] (released after submission deadline) also analyses the sensitivity of DMs to error propagation, and aims to bound this error by scaling the long skip-connections. While the method in [1] is sufficiently different to the proposed method, could the authors comments on this? It would appear that [1] also bounds the error, without the computational overhead.
2. The proposed method has significant computational overhead (Figure 4). How does the comparisons to  baselines (Table 1) change at equal wall-clock time?



[1] https://arxiv.org/abs/2310.13545

---

> ### Author Response · Authors · 2023-11-14
>
> Dear Reviewer hknr,
>
> We thank you for your kind and constructive feedback.
>
> ### Comment-1: Contemporaneous work[1] (released after submission deadline) also analyses the sensitivity of DMs to error propagation … could the authors comments on this? …
>
> Answer-1: Thanks for pointing out this interesting paper. The paper studied how the coefficient scales of skip connections affect the training stability of U-Net and introduced two new scaling schemes (one is predefined scales and the other is learnable ones) to better model the skip connections. For the experiment, the paper showed that their proposed methods stabilized and accelerated the training of U-Net.
>
> Your mentioned paper differs from our work in the following 3 points:
> 1) In terms of **research topic**, the paper focused on the training stability of U-Net (which is inside the denoising model $\epsilon_{\theta}$), while our work concentrates on the error accumulation of the backward process (which is outside the model);
> 2) In terms of **method**, the paper proposed to better scale the skip connections of U-Net (which makes its training more stable and faster), while our work introduces a regularization loss to reduce the error propagation of the backward process (which improves the generation quality);
> 3) In terms of **theory**, the paper developed theorems that estimated the feature norms and gradient magnitudes of U-Net (which is inside the model $\epsilon_{\theta}$), while our work builds a theoretical framework to analyze the error propagation of backward process (which is inside the model).
>
> For your comment “[1] also bounds the error”, note that **the error defined in that paper (i.e., the sensitivity of U-Net w.r.t. the input perturbation) is different from our defined errors** (i.e., the denoising error and its accumulation along the backward process). Regarding your concern about the computational overhead, our model indeed has a higher time cost for training, but the generation quality of diffusion models is also improved accordingly.
>
> We will cite your mentioned paper and include the above discussion in the revised version.
>
> ### Comment-2: … gains need to be weighed against this increase in compute. The proposed method has significant computational overhead (Figure 4). How does the comparisons to baselines (Table 1) change at equal wall-clock time?
>
> Answer-2: Thanks for raising this point. A reminder is that **the y-axis in Fig. 4 of our paper actually represents the time cost of $200$ training steps**, rather than just $1$ step. We apologize if this mistake has misled you to the impression that our method is very inefficient.
>
> As you suggested, **we have compared our model with the two most efficient baselines** (i.e., DDPM and DDIM) on CIFAR-10 with the same training time. For Consistent DM and FP-Diffusion, our model performs better than them and has a lower time cost per training step. The experiment results of FID scores and time costs are as follows.
>
> | Model                  | Training Steps | Total Training Time | FID Score |
> |------------------------|----------------|---------------------|-----------|
> | DDPM                   | 200K           | 10hrs               | 3.61      |
> | ADM-IP                 | 190K           | 10hrs               | 3.29      |
> | **DDPM w/ Our Method** | **98K**            | **10hrs**               | **3.16**      |
> | DDPM w/ Our Method     | 200K           | 20hrs               | 2.93      |
>
> From the above table, we can see that **our proposed method still outperforms the baselines with the same amount of training time**. For example, our method reduces the FID score of DDPM on CIFAR-10 by 12.47%, using no more than half of its training steps. Besides, while leading to a higher time cost per training step, **our method has no impact on the inference speed of diffusion models, which is more important in practical applications**.

---

### Official Review · Reviewer_e2Ed · 2023-10-31

**Soundness:** 3 good
**Presentation:** 3 good
**Contribution:** 3 good
**Rating:** 6
**Confidence:** 4

**Summary:**

This work analyze the error propagation of diffusion models by introducing the modular error (KL divergence between the reverse conditional distribution at each step), the cumulative error (KL divergence between the marginal distribution at each step). And then introduce an regularization loss based on MMD estimation for the cumulative loss to reduce the cumulative error and improve the sample quality of the trained diffusion model.

**Strengths:**

- The proposed method is easy to understand and the writing is clean and easy to follow.
- The error propagation of diffusion models is an important question and worth to be studied. The topic is important.

**Weaknesses:**

Major:

- The assumption in the core theorem is absolutely wrong.
  - "suppose that the output of neural network $\epsilon_\theta$ follows a standard Gaussian", which cannot be true. Because the noise-pred model corresponds to the denoising score matching loss, it is proved that the ground truth of such model is propotional to the score function of the distribution, i.e., $\nabla_{x_t} \log q_t(x_t)$. For a small $t$, such score function is quite complex and cannot be a simple and single-mode Gaussian distribution, and is far different.
- Remark 3.2 is not rigorous. The proof requires $T$ goes to infty, but it is not true in practice.
- Lack of detailed settings of experiments: what is the sampling algorithm for obtaining the FID results? What is the detailed network structure (e.g., layer structure and number of hidden neurons) and amount of parameters?

**Questions:**

1. Please address and fix the proof of the main theorem.

2. Please add more detailed descriptions of the experiment settings to ensure reproducibility.

=====================

I've carefully read the authors' responses to other reviewers and AC. I deeply appreciate the authors' efforts to address the concerns, and now I don't have more questions. I think it is a quite interesting and useful technique for improving the training procedure of diffusion models, with the validation of the fine-tuning experiment results and the EDM results. So I raise the score to 6.

---

> ### Author Response · Authors · 2023-11-14
> **Part-1 of Our Response**
>
> Dear Reviewer e2Ed,
>
> Thanks for your review and the constructive feedback.
>
> In the following, we have answered all your concerns in a point-by-point manner. Importantly, **to address your concern about the two preconditions of our theorems, we introduce alternative assumptions that are much weaker but still make the theorems hold**. Our new assumptions are also very solid because we derive them from a theoretical angle and find support from either empirical experiments or existing works [1,2].
>
> ### Comment-1: … “suppose that the output of neural network $\epsilon_{\theta}$ follows a standard Gaussian", which cannot be true. Because the noise-pred model corresponds to the denoising score matching …
>
> Answer-1: Thanks for raising this point. In this answer, **we first explain the role of your mentioned assumption in Theorem 3.1 and then replace it with a much weaker assumption**, which is derived from a theoretical perspective and still makes the theorem hold. **Lastly, we perform experiments to show that our new assumption is indeed valid in practice**.
>
> Since neural network $\epsilon_{\theta}(x_t, t)$ is tasked to fit Gaussian noise $\epsilon$ in the loss function $L^{nll}$, we previously assumed that the output distribution of $\epsilon_{\theta}(x_t, t)$ follows a standard Gaussian. In our proof to Theorem 3.1, **that assumption is applied to indicate that $E[\| \epsilon_{\theta}(x_t, t) \|^2] = K$** ($K$ is the vector dimension). However, considering Eq. (27) and Eq. (28) in the proof, **our theorem still holds for $E[\| \epsilon_{\theta}(x_t, t) \|^2] \ge K$**.
>
> To derive a new assumption, we first set a term
>
> $r_t = E[ \| \epsilon - \epsilon_{\theta}(\sqrt{\bar{\alpha}_t} x_0 + \sqrt{1 - \bar{\alpha}_t} \epsilon, t) \|^2 ]$,
>
> which indicates the prediction error of neural network $\epsilon_{\theta}(x_t, t)$. The term will vanish to $0$ if $\epsilon_{\theta}(x_t, t)$ is fully accurate in backward denoising. According to the Triangle Inequality, we then have
>
> $r_t \ge E[ \| \epsilon \|^2 - \| \epsilon_{\theta}(\cdot) \|^2  ] = E [\| \epsilon \|^2] - E[ \| \epsilon_{\theta}(\cdot) \|^2 ] $.
>
> Because the second moment of Gaussian distribution $E[\| \epsilon \|^2]$ is $K$, we further have
>
> $(1 / K) E[ \| \epsilon_{\theta}(\sqrt{\bar{\alpha}_t} x_0 + \sqrt{1 - \bar{\alpha}_t} \epsilon, t) \|^2 ] \ge 1 - (r_t / K) $.
>
> Finally, consider the fact [1] that $x_t$ can be reparameterized as $\sqrt{\bar{\alpha}_t} x_0 + \sqrt{1 - \bar{\alpha}_t} \epsilon, \epsilon \sim N(0, I) $ and let the prediction error $r_t$ vanish, **the above equation motivates us to make the following new assumption**:
>
> $(1 / K) E[ \| \epsilon_{\theta}(x_t, t) \|^2 ] \ge 1$,
>
> **which is much weaker than the previous assumption but still makes our Theorem 3.1 hold**.
>
> We perform experiments on 2 datasets to verify our new assumption. For every dataset, we sample $10000$ trajectories $x_{0:T}$ from a well-trained diffusion model. In this process, we collect $10000$ outputs from neural network $\epsilon_{\theta}(x_t, t)$ at every step $t$ and use them to estimate $(1 / K) E[ \| \epsilon_{\theta}(x_t, t) \|^2 ]$. Below are the experiment results.
>
> | Dataset | t=1000 | t=800 | t=600 | t=400 | t=200 | t=1   |
> |---------|--------|-------|-------|-------|-------|-------|
> | CIFAR-10 | 1.17   | 1.19  | 1.15  | 1.11  | 1.08  | 1.05  |
> | CelebA  | 1.25   | 1.21  | 1.23  | 1.17  | 1.11  | 1.09  |
>
> From the above table, we can see that **the estimated value of $(1 / K) E[ \| \epsilon_{\theta}(x_t, t) \|^2 ]$ is consistently larger than $1$ at every backward iteration for both datasets**. Therefore, our new assumption $(1 / K) E[ \| \epsilon_{\theta}(x_t, t) \|^2 ] \ge 1$ has strong empirical support.

---

> ### Author Response · Authors · 2023-11-14
> **Part-2 of Our Response**
>
> ### Comment-2: Remark 3.2 is not rigorous. The proof requires $T$ goes to infty, but it is not true in practice.
>
> Answer-2: Thanks for pointing out this. We address your concern by **introducing an alternative precondition that is weaker than the previous one but still makes our proposition hold**. Importantly, our new precondition is supported by current works [1,2].
>
>  In our proof to Proposition A.1, the use of that precondition $T \rightarrow \infty$ is to let $q(x_T | x_0) \rightarrow N(x_T; 0, I)$. Considering the fact [1] that
>
> $q(x_t | x_0) = N(x_t; \sqrt{\bar{\alpha}_t} x_0, \sqrt{1 - \bar{\alpha}_t} \epsilon)$,
>
> **we can achieve the goal with a much weaker assumption: $\lim_{t \rightarrow T} \bar{\alpha_t} \rightarrow 0$**. Notably, **this new assumption is a standard configuration in current diffusion models** (e.g., DDPM [1] and SGM [2]), which lets $x_T$ contain no information about $x_0$.
>
> ### Comment-3: Lack of detailed settings of experiments: what is the sampling algorithm for obtaining the FID results? What is the detailed network structure (e.g., layer structure and number of hidden neurons) and amount of parameters?
>
> Answer-3: For computing the FID scores, we adopt the same sampling procedure as DDPM (see Algo. 2 of [1]) and generate $50000$ samples with $1000$ backward iterations. For the network structure, we typically use U-Net with $4$ layers, $192$ channels, and channel multipliers as $\\{1,2,3,4\\}$, leading to a model size of 300M. We use Adam as the optimization algorithm with a learning rate of $10^{-4}$ and set the dropout ratio as $0.2$.
>
> Sec. 6.1 of our paper has some other details of the experiment setup. We will merge this answer into that section in the revised version.
>
> ## References
>
> [1] Ho et al., Denoising Diffusion Probabilistic Models, NeurlPS-2020.
>
> [2] Song et al, Score-Based Generative Modeling through Stochastic Differential Equations, ICLR-2021.

---

> ### Author Response · Authors · 2023-11-20
>
> Dear Reviewer e2Ed,
>
> We thank you for your time in reviewing our paper. With only 2 days left, we would like to know whether our previous response has addressed your concerns. Looking forward to your feedback!
>
> Best regards,
>
> The Authors

---

> > ### Comment · Reviewer_e2Ed · 2023-11-22
> > **Thanks for the rebuttal!**
> >
> > I've carefully read the authors' responses to other reviewers and AC. I deeply appreciate the authors' efforts to address the concerns, and now I don't have more questions. I think it is a quite interesting and useful technique for improving the training procedure of diffusion models, with the validation of the fine-tuning experiment results and the EDM results. So I raise the score to 6.

---

> ### Author Response · Authors · 2023-11-22
>
> Thank you, and thanks for your constructive review!

---

> ### Comment · Reviewer_hknr · 2023-11-22
> **Regarding gaussian distribution of gradients**
>
> Perhaps the authors could simply measure the gradients, and compare its histogram's R2 to the best fit gaussian?
> That may provide an alternative path to using the gaussian assumption.

---

> > ### Author Response · Authors · 2023-11-22
> >
> > Thanks for your kind suggestion. We did consider this way, but we finally weakened that assumption to make our theorem more solid.

---

### Official Review · Reviewer_MiNJ · 2023-11-01

**Soundness:** 3 good
**Presentation:** 2 fair
**Contribution:** 3 good
**Rating:** 8
**Confidence:** 2

**Summary:**

In this paper, the authors investigate error propagation in diffusion models. They develop a framework to define error propagation for diffusion and connect error propagation to generation quality. This enables them to use the measured error as a regularization term during the diffusion model training, improving the generation results of models in small-scale experiments.

**Strengths:**

- The proposed method is a novel approach to measuring error propagation in diffusion models and offers a new perspective on diffusion model training. The authors argue that apart from making the denoiser network more accurate, which has been the main focus of the literature so far, it is also important to regularize such that the denoiser is also robust to errors in the input during inference. This could have significant impacts on the broader diffusion generative model community.

- The presented methodology is principled and well-explained. The experiments clearly demonstrate the success of the proposed solution in mitigating error propagation in diffusion models.

**Weaknesses:**

- The authors briefly address the trade-off between increased training time and reduced error propagation (resulting in better FID) for the 32x32 images of CIFAR and ImageNet but do not mention their CelebA experiments on 64x64 images. It is not clear if the benefits scale with the image sizes without an increased overhead as it is possible that the error estimate requires more samples or a larger sampling length $L$.

**Questions:**

- Would it be possible to fine-tune pre-trained diffusion models with the regularization term to mitigate this error propagation a-posteriori? If the training time of adding the regularization makes training larger models prohibitive it would be interesting to explore whether it is possible to tune the denoiser network after having trained with just the $L^{nll}$ loss.

---

> ### Author Response · Authors · 2023-11-14
>
> Dear Reviewer MiNJ,
>
> Thank you for your kind and constructive comments. In particular, the idea of fine-tuning a trained diffusion model with our proposed regularization is very interesting and useful, which can potentially reduce the training time of our model. We have conducted experiments to show that your idea indeed works in practice.
>
> ### Comment-1: … do not mention their CelebA experiments on 64x64 images … It is not clear if the benefits scale with the image sizes without an increased overhead …
>
> Answer-1: Thanks for pointing out this. As a reminder, **the y-axis in Fig. 4 of our paper in fact represents the time cost of $200$ training steps** (not just $1$ step). We apologize if this mistake might mislead you to the impression that our method is very inefficient.
>
> To address your concern, we have performed an additional trade-off study (showing how the FID score and the time cost change w.r.t. increasing bootstrapping steps $L$) on CelebA. The results are in the following:
>
> | Bootstrapping Steps $L$ | 2      | 3      | 4      | 5      | 6      | 7      | 8      |
> |-------------------------|--------|--------|--------|--------|--------|--------|--------|
> | FID Score                  | 1.51   | 1.39   | 1.29   | 1.22   | 1.18   | 1.16   | 1.15   |
> | **Decrement of FID Score**  | N.A.   | 0.12   | 0.10   | 0.07   | 0.04   | 0.02   | 0.01   |
> | Time Cost Per Training Step | 0.37s | 0.45s | 0.55s | 0.63s | 0.72s | 0.81s | 0.92s |
>
> From the above table, we can see the FID scores tend to converge after $L=7$, which is similar to the situations of CIFAR-10 and ImageNet. **The results imply that the performance improvements contributed by our proposed regularization scales with the image size**. Importantly, while our method incurs an extra time cost per training step, it has no impact on the sampling speed of diffusion models, which is more important in practice.
>
> ### Comment-2: Would it be possible to fine-tune pre-trained diffusion models with the regularization term … it would be interesting to explore whether it is possible to tune the denoiser network …
>
> Answer-2: Thanks for providing such an interesting idea. To verify the feasibility of your idea, we have trained a diffusion model on CIFAR-10 in a way that **only the last 10% training steps involve our proposed regularization**. The results are as below.
>
> | Model                   | All Training Steps | Steps with Regularization | Training Time | FID Score |
> |-------------------------|---------------------------|---------------------------|----------------|-----------|
> | DDPM                    | 200K |0                         | 10hr           | 3.61      |
> | **DDPM w/ Our Proposed Reg.** | **200K** | **20K**                     | **11hr**          | **3.21**      |
> | DDPM w/ Our Proposed Reg.      | 200K |200K                      | 20hr           | 2.93      |
>
>
> From the above table, we can see that the diffusion model is still improved much (i.e., a reduction of the FID score by 11.08%) by partly applying our method, with a minor increase in training time. Therefore, **your idea indeed works empirically and is very useful in improving the efficiency of our proposed regularization**.
>
> We will include the above experiment and discussion in the revised version.

---

> > ### Comment · Reviewer_MiNJ · 2023-11-20
> >
> > I would like to thank the authors for taking the time to address my comments. I would strongly encourage you to include the fine-tuning results in the main text as it significantly helps with the presentation of the method. Given that my score is already high, I will be maintaining my rating.

---

> > > ### Author Response · Authors · 2023-11-20
> > >
> > > Sure, we will include your suggestion and the new experiment results in the revised version. Thank you and thanks again for your constructive review.

---

### Comment · Area_Chair_HgRi · 2023-11-21

My examination of the reviews and rebuttal indicates that assumptions were indeed made regarding the optimized denoising network output that may not hold true. The authors have tried to address this concern, as highlighted by Reviewer e2Ed, by weakening the assumptions.

## Reviewer e2Ed:

Given that only two reviewers expressed clear confidence in evaluating the merits of the paper and you were the sole reviewer to identify the aforementioned issue, I strongly encourage your participation in responding to the authors and engaging in discussions with the other reviewers and AC.

## Authors:

Q1: Have you tested your algorithm on a more recent diffusion codebase, such as EDM (https://github.com/NVlabs/edm), which has demonstrated superior performance compared to your lowest FID reported on CIFAR 10, using a significantly fewer number of reverse diffusion steps? I believe conducting these additional experiments is crucial to instill confidence in the reader that your method not only addresses the limitations of the DDPM codebase, but is also generalizable to diffusion models as a whole.

Q2: Utilizing MMD to align $p_{\theta}(x_{t-1})$ and $q(x_{t-1})$ appears computationally expensive. Have you encountered challenges in tuning the MMD kernel? Additionally, have you considered exploring alternative distribution matching methods, such as VAE and GAN? I would also suggest establishing connections with hybrid models like VAE+Diffusion and GAN+Diffusion, as they share similarities in the need to match $p_{\theta}(x_{t-1})$ and $q(x_{t-1})$.

---

> ### Author Response · Authors · 2023-11-21
>
> Dear AC,
>
> We thank you for managing the review process and for your help in improving our paper. We have also answered your questions in the following.
>
> ## Response to Q1
>
> Thanks for raising this point. As you suggested, we have conducted an experiment with the codebase of EDM. Below are the results for $T=35$ backward iterations.
>
> | Method                           | Result Source      | FID score on CIFAR-10 |
> | -------------------------------- | ------------------- | ---------------------- |
> | VE SDE w/ EDM                     | Table 2 in EDM [1] | 1.98                   |
> | VE SDE w/ EDM                     | Our Experiment     | 2.16                   |
> | **VE SDE w/ EDM, Our Method**         | **Our Experiment**    | **1.86**                   |
>
> The F1 score of our trained baseline (i.e., VE SDE w/ EDM) is very close to the result reported in the original paper [1] and our proposed regularization improves this score by 13.89%. The results indicate that our method not only improves DDPM but also is applicable to diffusion models in general.
>
> Thank you for helping us further highlight the merits of our proposed method.
>
> ## Response to Q2
>
> Thanks for your questions. MMD is much more efficient after applying our proposed bootstrapping trick (see Sec. 5.2 of our paper). Importantly, as we highlighted in the response to Reviewer hknr, our method still significantly improved the diffusion models and outperformed previous baselines with the same training time. Reviewer MiNJ also provided a very interesting idea to make our method more efficient: only applying our regularization for the last 10% training steps. Our experiments showed that this idea indeed works and we are very grateful to the reviewer.
>
> We did consider using the adversarial loss (i.e., GAN) to align $p_{\theta}(x_t)$ with $q(x_t)$. However, we finally chose MMD instead for the following two reasons:
> 1) GANs are known for their training instability  [2] and their sensitivity to hyper-parameter selection [3]. In contrast, MMD is parameter-free and more stable;
> 2) As proved in our paper (i.e., Proposition 4.1), MMD tightly bounds our defined cumulative loss from below and above. By using MMD, our theory and method are connected.
>
> To our knowledge, DDGAN [4] and Latent Diffusion [5] are two popular works that respectively combine diffusion models with GAN and VAE. However, they use GAN or VAE either as the denoising model or to pre-compress the training data. Compared with these models, our proposed regularization mainly aims to minimize the cumulative error (see Def. 3.2 of our paper), reducing error propagation along the backward process.
>
> Once again, we thank the AC and all the reviewers for all their helpful suggestions.
>
> ## References
> [1] Karras et al., Elucidating the Design Space of Diffusion-Based Generative Models, NeurIPS-2022.
>
> [2] Arjovsky and Bottou, Towards Principled Methods for Training Generative Adversarial Networks, ICLR-2017.
>
> [3] Lucic et al., Are GANs Created Equal? A Large-Scale Study, NeurIPS-2018.
>
> [4] Xiao et al., Tackling the Generative Learning Trilemma with Denoising Diffusion GANs, ICLR-2022.
>
> [5] Rombach et al., High-Resolution Image Synthesis with Latent Diffusion Models, CVPR-2022.

---

### Public Comment · ~Shikun_Sun1 · 2024-06-05
**About Gaussian Assumption**

Thank you very much for your excellent work!

I noticed the strong Gaussian assumption in the paper (Remark 3.3) and saw that you improved the proof during the rebuttal phase.

Could you please merge these updates into the PDF file?

Thank you again!

---

> ### Public Comment · ~Yangming_Li1 · 2024-06-05
>
> Dear Shikun,
>
> Thanks for your comment.
>
> We indeed did. Please refer to Appendix D.
>
> Best wishes,
>
> Yangming

---

### Meta-Review · Area_Chair_HgRi · 2023-12-08

**Metareview:**

To enhance the training and sampling processes of diffusion models (DMs), the paper explores error propagation in reverse diffusion and presents a framework to quantify and address this challenge. The authors introduce a regularization term aimed at mitigating error propagation, showcasing its effectiveness in refining DMs across three different low-resoltion image datasets. The study underscores the significance of comprehending and minimizing error propagation to elevate the sample quality in trained diffusion models.

**Justification For Why Not Higher Score:**

The paper's score may not be higher due to initial critical errors in the theoretical analysis, which might have impacted the overall credibility and robustness of the presented findings. Additionally, the experimental design initially relied on the DDPM sampler with 1000 sampling steps, and although the authors addressed concerns by including results on CIFAR10 with EDM and $T=35$ reverse steps, there still exists a level of concern regarding the generalizability of the proposed method to diffusion models as a whole. The need for more comprehensive evaluations is highlighted, suggesting that further testing and analysis could provide a clearer understanding of the method's performance and applicability.

**Justification For Why Not Lower Score:**

The score is not lower because all reviewers recommended acceptance. Although the AC has concerns about the generalizability of the observed benefits, s/he is willing to give the paper the benefit of the doubt.

---

### Decision · Program_Chairs · 2024-01-16

Accept (poster)